# A ten-year global monthly averaged terrestrial NEE inferred from the ACOS GOSAT v9 XCO$_2$ retrievals (GCAS2021)

Fei Jiang[1,2,4], Weimin Ju[1,2], Wei He[1], Mousong Wu[1], Hengmao Wang[1], Jun Wang[1], Mengwei Jia[1], Shuzhuang Feng[1], Lingyu Zhang[1], Jing M. Chen[1,3]

[1]Jiangsu Provincial Key Laboratory of Geographic Information Science and Technology, International Institute for Earth System Science, Nanjing University, Nanjing, 210023, China

[2]Jiangsu Center for Collaborative Innovation in Geographical Information Resource Development and Application, Nanjing, 210023, China

[3]Department of Geography, University of Toronto, Toronto, Ontario M5S3G3, Canada

[4]Frontiers Science Center for Critical Earth Material Cycling, Nanjing University, Nanjing, 210023, China

*Correspondence to*: Fei Jiang (jiangf@nju.edu.cn)

**Abstract.** A global gridded Net Ecosystem Exchange (NEE) of CO$_2$ dataset is vital in global and regional carbon cycle studies. Top-down atmospheric inversion is one of the major methods to estimate the global NEE, however, the existing global NEE datasets generated through inversion from conventional CO$_2$ observations have large uncertainties in places where observational data are sparse. Here, by assimilating the GOSAT ACOS v9 XCO$_2$ product, we generate a ten-year (2010–2019) global monthly terrestrial NEE dataset using the Global Carbon Assimilation System, version 2 (GCASv2), which is named as GCAS2021. It includes gridded (1°×1°), globally, latitudinally, and regionally aggregated prior and posterior NEE and ocean (OCN) fluxes, and prescribed wildfire (FIRE) and fossil fuel and cement (FFC) carbon emissions. Globally, the decadal mean NEE is -3.73±0.52 PgC yr$^{-1}$, with interannual amplitude of 2.73 PgC yr$^{-1}$. Combining the OCN flux, and FIRE and FFC emissions, the net biosphere flux (NBE) and atmospheric growth rate (AGR) as well as their inter-annual variabilities (IAVs) agree well with the estimates of Global Carbon Budget 2020. Regionally, our dateset shows that eastern North America, Amazon, Congo Basin, Europe, boreal forests, southern China and Southeast Asia are carbon sinks, while western US, African grasslands, Brazilian plateaus and parts of South Asia are carbon sources. In the TRANSCOM land regions, the NBEs of temperate N. America, northern Africa and boreal Asia are between the estimates of CMS-Flux NBE 2020 and CT2019B, and those in temperate Asia, Europe, and Southeast Asia are consistent with CMS-Flux NBE 2020 but significantly different from CT2019B. In the RECCAP2 regions, except for Africa and South Asia, the NBEs are comparable with the latest bottom-up estimate of Ciais et al. (2021). Compared with previous studies, the IAVs and seasonal cycles of NEE of this dataset could clearly reflect the impacts of extreme climates and large-scale climate anomalies on the carbon flux. The evaluations also show that the posterior CO$_2$ concentrations at remote sites and in regional scale, as well as on vertical CO$_2$ profiles in the Asia-Pacific region, are all consistent with independent CO$_2$ measurements from surface flask and aircraft CO$_2$ observations, indicating that this dataset captures surface carbon fluxes well. We believe that this dataset can contribute to regional or national-scale carbon cycle and carbon neutrality assessment, and carbon dynamics research. The dataset can be accessed at

https://doi.org/10.5281/zenodo.5829774 (Jiang, 2022).

## 1 Introduction

35    Terrestrial ecosystem uptakes $CO_2$ from the atmosphere through photosynthesis and releases $CO_2$ into the atmosphere through respiration. Its net carbon exchange (NEE) plays a very important role in regulating the atmospheric $CO_2$ concentration, thereby slowing down the global warming. However, NEE has significant spatial differences and inter-annual variations (IAV) (Bousquet et al., 2000; Piao et al., 2020). Therefore, accurately quantifying global and regional NEE and clarifying their drivers of IAV is a key scientific issue in global carbon cycle research, and a reliable global NEE dataset is vital for this research.

40    Until now, a series of global NEE or net biosphere exchange (NBE = NEE + wildfire carbon emission) products like FLUXCOM (Jung et al., 2009), TRENDY (Sitch et al., 2015), Jena CarboScope (Rödenbeck et al., 2003), CT2019B (Jacobson et al., 2020), and CMS-Flux NBE 2020 (Liu et al., 2021), are available and widely used in different studies, which were created using data-driven machine learning methods, ecosystem models, or inversion models. Machine learning methods estimate global carbon flux by upscaling eddy covariance data (Zeng et al., 2020), ecosystem models simulate photosynthesis and respiration of ecosystems based on meteorological, soil, and land cover data and a series of parameters (Chen et al., 1999), and inversion models estimate surface $CO_2$ fluxes using the globally distributed atmospheric $CO_2$ observations and/or satellite retrievals of column averaged $CO_2$ dry air mole fraction ($XCO_2$) (Enting and Newsam, 1990; Gurney et al., 2002; Jiang et al., 2021). Different methods have their own advantages and disadvantages. The NEE estimated by top-down atmospheric inversions is determined by the density and accuracy of the $CO_2$ observations, the accuracy of modeled atmospheric transport, and knowledge of the prior uncertainties of the flux inventories (Liu et al., 2021). Generally, in situ and flask $CO_2$ observations have high precision, with measurement error lower than 0.2 ppm, however, the global distribution of flask or in-situ sites is extremely uneven, there are many sites over North America (N. America) and Europe, but very few sites over tropics, Africa, and southern oceans (Schuldt et al., 2020). Therefore, the inversions generally have robust performance on global or hemisphere scale (Houweling et al., 2015), but on regional scales, due to the uneven distribution of observations, the reliability of inversion results varies greatly in different regions (Peylin el al., 2013).

55    Satellite $XCO_2$ retrievals from the Greenhouse Gases Observing Satellite (GOSAT) (Kuze et al., 2009) and the Observing Carbon Observatory 2 (OCO-2) (Crisp et al., 2017) have much better spatial coverage (O'Dell et al., 2018) than ground-based observations. Although the accuracy of $XCO_2$ is relatively lower (~ 1 ppm, Kulawik et al., 2019) compared to flask and in-situ observations, and the response of $XCO_2$ to changes in the surface carbon flux is weaker, many inversion studies have proved that satellite $XCO_2$ retrievals could improve the estimates of surface carbon fluxes (e.g., Basu et al., 2013; Maksyutov et al., 2013; Saeki et al., 2013; Chevallier et al., 2014; Deng et al, 2016), especially for the fluxes in Africa, South America (S. America), and Asia, where the sparsity of the surface monitoring sites is most evident (Takagi et al., 2011). Wang et al. (2019)

compared the NEE inferred from GOSAT and OCO-2 retrievals, and surface flask observations, and found that the performance of inversion with GOSAT data only was comparable with the one using surface observations. Moreover, studies also showed that with satellite $XCO_2$ retrievals, the inverted carbon flux could well reveal the impact of extreme droughts and large-scale climate anomalies on regional and continental terrestrial carbon dynamics (Liu et al., 2018; Deng et al, 2016; Detmers et al., 2015; Jiang et al., 2021).

By assimilating both GOSAT and OCO-2 $XCO_2$ retrievals, Liu et al. (2021) generated a global gridded monthly NBE product (i.e., CMS-Flux NBE 2020) using the NASA Carbon Monitoring System Flux (CMS-Flux) inversion framework (Liu et al., 2014, 2017, 2018; Bowman et al., 2017). This dataset spans over 2010–2018, in which the data from 2010-2014 and 2015-2018 were inferred from GOSAT $XCO_2$ and OCO-2 data, respectively. GOSAT and OCO-2 $XCO_2$ have large differences on spatial resolution and coverage, which may lead to discontinuities in the inversion results of certain regions. The ACOS GOSAT v9 $XCO_2$ data is now available on the NASA Goddard Earth Science Data and Information Services Center (GES-DISC), which spans from April 2009 to June 2020, and has been well bias corrected and quality filtered (Taylor et al., 2021).

In this study, based on the GOSAT v9 $XCO_2$ retrievals, we generate a 10-year global monthly NEE dataset from 2010 to 2019 (GCAS2021) using a well-constructed Global Carbon Assimilation System, version 2 (GCASv2) (Jiang et al., 2021; Wang et al., 2021a). Different from Liu et al. (2021), GCAS2021 focuses on NEE, because the wildfire (FIRE) emission was not optimized in this study. The optimized ocean flux and prescribed FIRE and fossil fuel and cement carbon (FFC) emissions are also included in this dataset. Users who want to use NBE data, could combine the NEE and FIRE emission by themselves. It is worth pointing out that since we have not optimized FIRE emissions, the optimized NEE may include compensation for the errors in FIRE emissions. This manuscript is organized as follows: Section 2 details the GOSAT retrievals, prior fluxes, and the GCASv2 system as well as uncertainty settings. Section 3 introduces the evaluation data and method, Section 4 briefly describes the dataset, Section 5 presents the characteristics of the dataset, including the estimates of global carbon budget and regional NEE as well as their IAVs, Section 6 details the evaluations results against independent $CO_2$ observations, and Section 7 gives a summary and the main conclusions.

## 2 Methods and data

### 2.1 The ACOS v9 GOSAT $XCO_2$ retrievals

The GOSAT satellite launched in 2009 (Kuze et al., 2009) was developed jointly by the National Institute for Environmental Studies (NIES), the Japanese Space Agency (JAXA) and the Ministry of the Environment (MOE) of Japan, which was designed to retrieve total column abundances of $CO_2$ and $CH_4$. In this study, the GOSAT $XCO_2$ retrieval is the ACOS Version 9.0 Level 2 Lite product (Taylor et al., 2021) at the pixel level during May 2009 - Dec 2019. The bias correction and quality filtering of this $XCO_2$ product have been evaluated using estimates derived from the Total Carbon Column Observing Network (TCCON)

as well as values simulated from a suite of global atmospheric inverse modeling systems (models), the results show that the differences in $XCO_2$ between GOSAT v9 and both TCCON and models have an one sigma error of approximately 1 ppm for ocean-glint observations and 1 to 1.5 ppm for land observations, and globally, the mean biases are less than approximately 0.2 ppm (Taylor et al., 2021). Compared with its previous version (ACOS v7.3), the proportion of data with a 'good' $XCO_2$ quality flag has increased from 3.9 % in v7.3 to 5.4% in v9.

The GOSAT $XCO_2$ retrievals have a resolution of 10.5 $km^2$ at nadir. Considering the facts that the resolution of a global atmospheric transport model is significantly lower than that of $XCO_2$ retrievals, we re-grid the $XCO_2$ data into 1°×1° grid cells. The pixel level $XCO_2$ data are filtered with xco2_quality_flag, which is a simple quality flag denoting science quality data (0=Good, 1=Bad), and provided along with the $XCO_2$ product. In each 1°×1° grid and each day, only the $XCO_2$ with xco2_quality_flag equals 0 are selected and averaged according to Equation (1).

$$C_{G,T} = \frac{1}{W}\sum_{l=1}^{W} C_{l,t} , \quad T = \frac{1}{W}\sum_{l=1}^{W} t \quad\quad\quad (1)$$

where $C_{l,t}$ denotes the selected pixel level $XCO_2$ located in 1°×1° grid $G$ of one day, $l$ is the identifier of the record, $t$ is the observation time, and $W$ denotes the number of $C_{l,t}$. $T$ is the averaged observation time, and $C_{G,T}$ is the re-grided $XCO_2$ concentrations. The other variables in the $XCO_2$ product like column-averaging kernel, retrieval error, etc., which will be used in the calculations of simulated $XCO_2$, are also re-grided using this method.

## 2.2 Prior $CO_2$ fluxes

The prior carbon fluxes used in this study consist of terrestrial NEE, FIRE carbon emission, FFC carbon emission, and $CO_2$ exchanges over the ocean surface (OCN). NEE in 3-hour interval is simulated using the Boreal Ecosystems Productivity Simulator (BEPS) model, details about the BEPS simulations please refer to Chen et al. (2019). FIRE emission is directly obtained from the Global Fire Emissions Database, Version 4.1 (GFED4s) (van der Werf et al., 2017; Mu et al., 2011). FFC emission is an average of two products from Carbon Dioxide Information Analysis Center (CDIAC) (Andres et al., 2011) and Open-source Data Inventory of Anthropogenic $CO_2$ (ODIAC) (Oda et al., 2018), respectively. OCN flux is derived from the Takahashi et al. (2009) climatology of seawater $pCO_2$. Both FFC emission and OCN flux were downloaded from CT2019B (Jacobson et al., 2020). It should be noted that there are no data in the $pCO_2$-Clim product in many offshore areas like Japan Sea, Mediterranean, Gulf of Mexico, and East China Sea. Following Jiang et al. (2021), the fluxes in 2009 modeled using a combined global ocean circulation (OPA) and biogeochemistry model (PISCES-T) (Buitenhuis et al., 2006) is used to fill the no data areas. The sea-air $CO_2$ fluxes simulated using the PISCES-T model have been used in many studies of ocean carbon cycle dynamics (e.g., McKinley et al., 2006; Valsala et al., 2012; Le Quéré et al., 2007), and also used as a priori ocean fluxes in previous inversion studies (e.g., Jiang et al., 2014; Deng et al., 2011; Chen et al., 2017). In addition, the CT2019B product is only until the beginning of 2019. OCN flux in 2019 is assumed to be the same as 2018. FFC emission is adjusted from the

emission in 2018 by ratios of 2019/2018 in different countries or regions (Figure S1), which was calculated based on the 2018 and 2019 emissions compiled by the Global Carbon Budget 2020 (GCP2020, Friedlingstein et al., 2020).

**2.3 The Global Carbon Assimilation System (GCAS, version 2)**

The global monthly NEE dataset is inferred using the Global Carbon Assimilation System, version 2 (GCASv2), which was developed for estimating gridded surface carbon fluxes mainly using satellite $XCO_2$ retrievals (Jiang et al., 2021). In this system, the Model for Ozone and Related Chemical Tracers, version 4 (MOZART-4) (Emmons et al., 2010) was coupled to simulate 3-D atmospheric $CO_2$ concentrations, and the Ensemble square root filter (EnSRF) algorithm (Whitaker and Hamill, 2002) was used to implement the inversion of surface fluxes. GCASv2 runs cyclically, and in each cycle (DA window), we use a "two-step" calculation scheme to maintain quality conservation. First, the prior fluxes are optimized using $XCO_2$ data, and then, the optimized fluxes are put again into the MOZART-4 model to generate the initial condition (IC) of the next window. In order to reduce the representative error of $XCO_2$, a 'super-observation' approach is also adopted, in which a super-observation is generated by averaging all observations located within the same model grid within a DA window; and to reduce the impact of spurious correlations, a localization technique is employed to determine which super-observations will be used for the current grid's optimization, which is based on the correlation coefficient between the simulated concentration ensembles in each observation location and the perturbed fluxes in current model grids, and their distances. For details, please refer to Jiang et al. (2021).

In this study, GCASv2 was run from May 1, 2009 to Dec 31, 2019 with the DA window of 1 week. The IC of 3-D $CO_2$ concentrations at 00:00 UTC May 1, 2009 was obtained from the product of CarbonTracker, version 2017 (CT2017). The first 8 months are considered as a spin-up run, and the results from Jan 1, 2010 to Dec 31, 2019 are analyzed and evaluated in this study. MOZART-4 is driven by the GEOS-5 meteorological fields, which has a spatial resolution of 1.9°×2.5°, and vertical level of 72 layers. MOZART-4 uses the same spatial resolution and the lowest 56 vertical levels of GEOS-5. Following Jiang et al. (2021), the model-data mismatch error of $XCO_2$ is constructed using the $XCO_2$ retrieval errors, which are provided along with the $XCO_2$ product and re-gridded using the same method as described in section 2.1. All retrieval errors are also uniformly inflated by a factor of 1.9 in this study, but a lowest error is fixed as 1 ppm.

There are four state vectors combining schemes in GCASv2, including 1) only the NEE is treated as state vector and optimized, 2) both NEE and OCN flux are state vectors; 3) NEE, OCN flux and FFC emissions are optimized at the same time; and 4) only net flux is optimized. In this study, the second scheme was selected, both NEE and ocean flux are optimized, and the FIRE and FFC are prescribed. The perturbation of prior fluxes is described in Equation (2), where $\delta_i$ represents random perturbation samples, and is drawn from Gaussian distributions with mean zero and standard deviation of one. $i$ is the identifier of the perturbed samples, N is the ensemble size (here 50). $\lambda$ is a set of scaling factors, which represents the uncertainty of

each prior flux. $X^b_{NEE}$, $X^b_{FIRE}$, $X^b_{FFC}$, and $X^b_{OCN}$ represent the prior fluxes of NEE, FIRE, FFC and OCN, respectively. The spatial resolution of the perturbation factor ($\delta_i \times \lambda$) we adopted is $3° \times 3°$, and the resolution of the prior fluxes is $1° \times 1°$, that is, the prior fluxes within each 3° grid have the same perturbation factor. In each 3° grid, $\lambda_{NEE}$ and $\lambda_{ocn}$ are set to be 6 and 10, respectively, which are corresponding to a global 1-$\delta$ uncertainty for NEE and OCN flux about 0.6 and 0.2 PgC yr$^{-1}$, respectively (for the method, see Text S1).

$$X^b_i = \lambda_{NEE} \times \delta_{i,NEE} \times X^b_{NEE} + \lambda_{ocn} \times \delta_{i,ocn} \times X^b_{OCN} + X^b_{Fire} + X^b_{FFC}, i = 1, 2, \ldots, N \qquad (2)$$

## 3 Evaluation data and method

Due to the huge difference of spatial scale between the inverted and directly observed fluxes, generally, it is impossible to directly validate the posterior NEE using observations, and instead, we indirectly evaluate the posterior flux by comparing the forward simulated atmospheric $CO_2$ mixing ratios against independent $CO_2$ measurements (e.g., Jiang et al., 2021; Wang et al., 2019; Feng et al., 2020). Therefore, a forward simulation using the MOZART-4 model and the posterior fluxes were conducted to create posterior $CO_2$ concentrations. For comparison, the prior $CO_2$ concentrations were also simulated with the prior fluxes. The simulation period, model configuration of MOZART-4 as well as initial field are the same as the assimilation experiment as described in section 2.3.

Surface flask and aircraft $CO_2$ observations are used for these independent evaluations in this study, which were obtained from the obspack_co2_1_GLOBALVIEWplus_v6.0_2020-09-11 product (OBSPACKv6, Schuldt et al., 2020). OBSPACKv6 contains a collection of discrete (flask), programmable flask package (PFP) and quasi-continuous (in-situ) measurements at surface, tower, ship and aircraft sites contributed by national and universities laboratories around the world. In this study, surface flask $CO_2$ measurements (including surface PFP) from 74 sites, and aircraft measurements (including flask, PFP and in-situ measurement methods) from 3 projects, are selected to evaluate the posterior $CO_2$ concentrations. There are 148 surface flask and PFP sites of observations in OBSPACKv6. The 74 sites were selected according the following processes: 1) only the sites with data more than 7 years during 2010 – 2019 were selected (48 sites removed); 2) for one location, if there are observations from different institutes, only the data provided by the NOAA Global Monitoring Laboratory (with lab number of 1 in each filename) were selected (21 sites removed); 3) for one location, if both flask and PFP observations are available, only flask observations were adopted (1 site removed); 4) for PFP site, if there are observations at different heights, only the observations at the top level were used (1 site removed); and 5) during the evaluations, we find that MOZART-4 model is unable to capture the variations of $CO_2$ mixing ratios at BKT and LJO, thus these site were also removed. The locations of the 74 sites are shown in Figure 1 and the corresponding sites code as well as the information about latitude and longitude are listed in Table S2 in the Supporting Information.

There are 76 aircraft observation sites (independent data files) in OBSPACKv6. In this study, we chose observations from the Comprehensive Observation Network for Trace gases by Airliner (CONTRAIL) project (Machida et al., 2008, 2018; Matsueda et al., 2008, 2015), the HIAPER Pole-to-Pole Observations (HIPPO) programme (Wofsy et al., 2011), and the lower-troposphere greenhouse-gas sampling programme in the Amazon basin of the CARBAM project (Gatti et al., 2014, 2021) to further evaluate the posterior $CO_2$ concentrations. The CONTRAIL project measures $CO_2$ concentrations using Continuous $CO_2$ Measuring Equipment (CME) on two passenger aircrafts (Boeing 747-400 and 777-200ER), thus there are observations along flight paths (including level flight, taking off and landing) from Japan to N. America, to Europe, to Hawaii, to Australia, and to Southeast and South Asia (Figure 2). During the taking off and landing, vertical profiles of $CO_2$ concentrations near airports were observed. As shown in Figure 1, there are few surface observations over the Asia-Pacific region, especially in Southeast and South Asia, therefore, the $CO_2$ vertical profiles near 8 cities over the Asia-Pacific region are selected in this study. The 8 cities are Hong Kong, Singapore, Jakarta, Bangkok, Sydney, New Delhi, Shanghai, and Tokyo. The HIPPO programme completed aircraft measurements spanning the Pacific from 85 ° N to 67 ° S during the periods of March to April 2010, and June to September 2011, with vertical profiles every approximately 2.2 ° of latitude (Wofsy et al., 2011). The CARBAM project conducted vertical $CO_2$ measurements at 4 sites (i.e., ALF, RBA, SAN, TAB, and TEF) in the Amazon basin during 2010 ~ 2018 (Figure 2) with small aircrafts and PFP equipment. TAB was from 2010 to 2012, and TEF started in 2013. During the evaluation of this study, TAB and TEF are combined as one site of TAB_TEF. At each site, 1-3 spiral profiles from approximately 4420 m to about 300 m a.s.l. were observed in each month. It is worth noting that OBSPACKv6 only provides ALF, RBA, SAN and TAB observations from 2010 to 2012, the rest data were downloaded from Gatti et al. (2021). For the CONTRAIL vertical profiles, the observations between the heights of 2 and 6 km are used, because the data measured below 2000 m are highly affected by local emissions (Jiang et al., 2014) due to the frequently ascending and descending of aircrafts. And for the HIPPO and CARBAM observations, the data above 1 km are adopted.

Four basic statistical measures, i.e., mean bias (BIAS), mean absolute error (MAE), root mean square error (RMSE), and correlation coefficient (CORR), are calculated against the surface and aircraft $CO_2$ observations, respectively. The functions of these 4 basic statistical measures are expressed as:

$$BIAS = \frac{1}{M}\sum_{j=1}^{M}(x_j - y_j) = \bar{y} - \bar{x} \tag{3}$$

$$MAE = \frac{1}{M}\sum_{j=1}^{M}|x_j - y_j| \tag{4}$$

$$RMSE = \sqrt{\frac{1}{M}\sum_{j=1}^{M}(x_j - y_j)^2} \tag{5}$$

$$CORR = \frac{\sum_{j=1}^{M}(x_j - \bar{x})(y_j - \bar{y})}{\sqrt{\sum_{j=1}^{M}(x_j - \bar{x})^2}\sqrt{\sum_{j=1}^{M}(y_j - \bar{y})^2}} \tag{6}$$

where $x_j$ and $y_j$ denote the modeled and the observational values, respectively, at the $j$th out of $M$ records, and the overbars denote averages. The BIAS, MAE, RMSE, and CORR reflect the overall model tendency, both the model bias and error variance, and the linear correspondence between the modeled and observational values, respectively.

## 4 Dataset description

GCAS2021 includes (1) monthly and annual prior and posterior NEE and OCN fluxes, and prescribed FIRE and FFC emissions in a global spatial resolution of 1°×1°; (2) globally, latitudinally, and regionally aggregated monthly and annual posterior NEE and NBE, and their uncertainties; and (3) weekly gridded ensemble members of posterior NEE and OCN fluxes. The regional fluxes are aggregated both in the TRANSCOM (Gurney et al., 2003) and the REgional Carbon Cycle Assessment and Processes Project (RECCAP2, Ciais et al., 2020) regions (Figure 3). The latitudinal fluxes are aggregated in northern mid-high latitudes (> 30° N, NL), tropical latitudes (30° S ~ 30° N, TL), and southern middle latitudes (<30° N, SL). The weekly gridded ensemble members could be used for calculating the posterior uncertainties based on user defined regional masks. We also provide a Fortran program for the calculation of posterior uncertainties. The method for calculating posterior uncertainties is given in the Text S1 in the Supporting Information. The gridded data are in NETCDF-3 format, while the regional aggregated data are in xlsx format.

## 5 Characteristics of the dataset

### 5.1 Global carbon budgets

Table 1 presents the year-by-year and decadal averaged posterior global carbon budgets during 2010 ~ 2019 of this study. The global annual NEE is in the range of -2.51±0.53 to -5.24±0.50 PgC yr$^{-1}$ (negative means absorbing $CO_2$ from the atmosphere, and positive means releasing $CO_2$ to the atmosphere). The year of 2011 has the largest land sink in the decade, while the year of 2016 has the weakest one, with interannual amplitude reaching 2.73 PgC yr$^{-1}$. On average, the decadal mean NEE is -3.73±0.52 PgC yr$^{-1}$. The OCN flux has an overall increase trend from 2010 to 2009, with a mean of -2.64±0.16 PgC yr$^{-1}$. Compared with the prior NEE (Figure S9l), the posterior NEEs increase significantly from 2010 to 2012, and decrease to varying degrees (in range of 0.15 to 1.15 PgC yr$^{-1}$) from 2015 to 2019. Table 1 also lists the estimates from the CMS-Flux (CMS-Flux NBE 2020, Liu et al., 2021) and CarbonTracker (CT2019B, Jacobson et al., 2020) systems. CMS-Flux NBE 2020 is a product for the period of 2010-2018, in which the results of 2010-2014 were inverted from the GOSAT $XCO_2$ v7.3, and the rests were inferred from the OCO-2 $XCO_2$ v9 retrievals. Both GOSAT and OCO-2 retrievals were from the ACOS team, created using the same retrieval algorithm and validated using the same strategy (Liu et al., 2021). CT2019B is a product inverted from global surface, tower and aircraft $CO_2$ measurements. CMS-Flux NBE 2020 only presented the NBE results, and the FIRE emission used in this study and CT2019B are also different. Therefore, this comparison focuses on NBE. In 2010

and 2014, our estimates are close to CT2019B and significantly lower than the estimates of CMS-Flux NBE 2020; in contrast, in 2011, 2012, 2013, 2016 and 2017, they are comparable to CMS-Flux NBE 2020 and higher than those of CT2019B. In 2015, it is higher than both. Moreover, Figure 4 presents a comparison between the estimates of this study and GCP2020 (Friedlingstein et al., 2020). There are large differences for the land-use and land-cover change (LULCC) carbon emissions between this study and GCP2020, we directly use the FIRE emission from GFED 4.1s as prescribed land-use emission, while GCP2020 uses an average of three bookkeeping models (Houghton et al., 2017; Hansis et al., 2015; Gasser et al., 2020), which account for changes in all carbon pools affected by LULCC. Therefore, we also compared the NBE between this study and GCP2020. For GCP2020, the NBE is the sum of NEE and LULCC emissions. Additionally, GCP2020 also reported atmospheric growth rate (AGR) of $CO_2$ in the atmosphere, which was estimated directly from atmospheric $CO_2$ concentration measurements provided by the NOAA Earth System Research Laboratory (Friedlingstein et al., 2020). Ideally, the inverted global net carbon flux (i.e., AGR) should agree with the observed AGR. As shown in Figure 4, the interannual changes of global NBE and AGR of this study match well with the estimates of GCP2020, with CORR of 0.75 and 0.88, BIAS (this study minus GCP2020) of 0.15 and 0.25 PgC yr$^{-1}$, and MAE of 0.51 and 0.40 PgC yr$^{-1}$, respectively. The difference in NBE between this study and GCP2020 is partly due to the imbalance item in GCP2020, especially in 2016. It also should be noted that in this study, the AGR in 2019 is higher than that in 2015, and significantly higher than the observed value, which is mainly due to the abnormally low carbon sink in the tropical latitudes (TL, 30° S ~ 30° N) in this year (Figure 7). The reason may be related to the biases in the GOSAT $XCO_2$ retrievals in TL. We analyze the monthly changes of GOSAT $XCO_2$ in 2015 and 2019, and compare them with the OCO-2 $XCO_2$ retrievals (OCO-2 v10). We find that after detrending, in TL, the GOSAT $XCO_2$ in 2019 is higher than that in 2015, while OCO-2 is the opposite (Figure S3). For the prior fluxes, the CORR, BIAS, and MAE of NBE and AGR compared against the GCP2020 estimates are 0.16 and 0.49, -0.51 and 0.09 PgC yr$^{-1}$, and 0.63 and 1.10 PgC yr$^{-1}$ (Figure S2). These indicate that the estimate of global carbon budgets has been significantly improved after constrained by the GOSAT retrievals.

**5.2 Annual NEE averaged from 2010-2019**

Figure 5 shows the distributions of the mean posterior annual NEE during 2010 - 2019. Carbon uptakes mainly occur over eastern N. America, Amazon, Congo Basin, Europe, boreal forests, southern China, and southeast Asia; and carbon releases mainly occur in western N. America (main western US), the East African and Ethiopian Plateaus and the Sahel region (mainly the grasslands in Africa), the Brazilian plateau, and parts of South Asia. Compared with the prior NEE, the land sinks in western N. America, most S. America, the grasslands in Africa, most East and South Asia, and eastern Siberia are decreased, while the sinks in eastern N. America, Europe, and western Siberia are significantly increased (Figure S4). In N. America, the distribution of NEE constrained with GOSAT $XCO_2$ exhibits a similar pattern to that of a recent regional inversion using surface $CO_2$ and $^{14}CO_2$ measurements, which also showed significant sources over western US and sinks over central and eastern US (Basu et

al., 2020). By using the Community Land Model (CLM5.0) and a Data Assimilation Research Testbed (DART) that assimilated with remotely sensed observations of leaf area and above-ground biomass, Raczka et al. (2021) simulated the NEE over western US and also found that there are large areas with carbon release. The western US is dominated by natural lands, which is particularly vulnerable to forest mortality from droughts, insect attacks, and wildfires, Ghimire et al. (2015) found large carbon release legacy from bark beetle outbreaks across western US. In addition, the ageing and decline of forest may be another reason for the carbon release in western US (Sleeter et al., 2018). The significant sources of NEE in the grasslands of Africa are consistent with previous top-down estimates based on satellite retrievals (Palmer et al., 2019) and surface $CO_2$ measurements (Valentini et al., 2014). Many observations based on the eddy covariance also reported carbon sources of NEE in the savanna grassland of West and South Africa (e.g., Veenendaal et al., 2004; Räsänen et al., 2017; Quansah et al., 2015). The significant increase of carbon release in the grasslands of Africa may be related to the underestimates of carbon emissions from small fires in GFED 4s. The FIRE emission in GFED 4s was estimated based on global burned area, which were from coarse spatial-resolution sensors. Ramo et al. (2021) showed that coarse sensers are unsuitable for detecting small fires that burn only a fraction of a satellite pixel, and pointed out that the FIRE emission of Africa in GFED 4s was underestimated by about 31% in 2016.

Table 2 lists the aggregated mean posterior annual NEE, NBE and FIRE emissions during the 1- years for the 11 TRANSCOM regions and the 10 RECCAP2 regions. Compared with the prior NEE, the absolute relative changes in most TRANSCOM regions are greater than 50% (Figure S5) after constrained with GOSAT data. In all regions, the aggregated posterior NEE are negative, indicating a carbon sink in each region. For the 11 TRANSCOM regions, we estimate that Europe has the strongest sink, followed by boreal Asia, tropical S. America, and northern Africa has the weakest sink. Among the 10 RECCAP2 regions, Russia's sink is the strongest, followed by N. America and Europe, and West Asia's sink is the weakest. It is worth noting that the Europe's NEE in the TRANSCOM region is twice that in RECCAP2. This is because the coverage of Europe is different in TRANSCOM and RECCAP2, the former includes the entire European continent, while the latter does not include European Russia.

Figure 6 shows a comparison between the results of this study and previous studies for both the TRANSCOM and RECCAP2 regions. For the TRANSCOM region, as shown in Figure 6a, in temperate N. America, northern Africa, boreal Asia, the estimates of this study are between the results of CMS-Flux NBE 2020 and CT2019B; in temperate Asia, Europe, and tropical Asia, our estimates are very close to CMS-Flux, but are significant differences with CT2019B, conversely, in Australia, our estimates are very consistent with CT2019B, but are significantly different from CMS-Flux. In tropical S. America, our result shows a strong carbon sink, which is consistent with previous mean annual biomass sink estimate of $-0.39 \pm 0.10$ PgC yr$^{-1}$ in Amazon during the 1980–2004 period based on repeated censuses at a widespread forest plot network (Phillips et al., 2009) and is roughly consistent with a regional inversion in a wet year of $-0.25$ PgC yr$^{-1}$ based on aircraft $CO_2$ measurements

(Gatti et al., 2014), while CMS-Flux NBE 2020 and CT2019B are both carbon sources. On the contrary, in temperate S. America, our result shows a weak carbon source, while the other two are both carbon sinks. In addition, in southern Africa, our estimate is also significantly different from them, we show strong carbon source, while CMS-Flux NBE 2020 and CT2019B show weak sink and source, respectively. The differences between this study and CMS-Flux NBE 2020 may be related to the different $XCO_2$ products used. As mentioned before, the NBE of CMS-Flux from 2010-2014 and 2015-2018 were inferred from GOSAT and OCO-2 products, respectively. In general, OCO-2 $XCO_2$ has much better spatial coverage than GOSAT $XCO_2$. Wang et al. (2019) pointed out that data amount is one of the most important factors affecting the inversion results, generally, in one region with more $XCO_2$ data, the carbon flux relative to the prior flux is changed more. Therefore, we conduct an additional comparison for the periods of 2010 to 2014 and 2015 to 2018, respectively, since in the first stage, the $XCO_2$ used in these two studies are almost the same (both GOSAT), while in the second stage, they are different. As shown in Figure S6, except for southern Africa, the difference between the two is significantly smaller in 2010-2014 than in 2015-2018, especially in temperate S. America, northern Africa, and Australia, confirming that the significant differences are mainly from the different $XCO_2$ products used in these two studies. In addition to $XCO_2$ data, the prior carbon flux can also have a significant impact on the inversion results (Philip et al., 2019). We further examine the prior and posterior NBE over southern Africa in these two studies, and find that the prior NBE used in these two systems are quite different (a strong sink in CMS-Flux, and a source in this study). In the first stage, the NBE changes ($\Delta_{NBE}$, a posteriori minus a priori) due to the GOSAT constraints are quite small in both studies (Figure S7), resulting in the large difference in the posterior NBE between these two studies, while in the second stage, because of the better spatial coverage of OCO-2 $XCO_2$, the $\Delta_{NBE}$ in CMS-Flux increase significantly, resulting in a shift of NBE from a priori strong sink to a posteriori medium source, thus reducing the difference of the posterior NBE in these two studies. We also find that there is also an increase in the $\Delta_{NBE}$ in this study, which may be related to the increase of GOSAT $XCO_2$ data from 2010 to 2019 (Taylor et al., 2022).

Based on inventory data of carbon-stock changes and satellite estimates of biomass changes where inventory data are missing, Ciais et al. (2021) gave a state-of-the-art estimate for the NBE of the RECCAP2 regions for the period of 2000-2009, which was calculated by taking the sum of the carbon-stock change and lateral carbon fluxes from crop and wood trade, and riverine-carbon export to the ocean. Figure 6b shows a comparison between this study and Ciais et al. (2021). Although the inverted NBE is not completely equivalent to the land sink obtained by the bottom-up method, generally, to reconcile top-down and bottom-up results, the inverted NBE should be adjusted with the lateral transport of reduced carbon compounds (RCC) and carbon release from net imported products (Ciais et al., 2008; Jiang et al., 2016). Overall, except for Africa and South Asia, the NBE estimated in this study and Ciais et al. (2021) are comparable. In Africa, we show a strong carbon source of $0.87\pm0.27$ PgC $yr^{-1}$, while Ciais et al. (2021) reported a very weak sink of $-0.07 \pm 0.29$ PgC $yr^{-1}$. Until now, there are still big differences in top-down estimates of African NBE in different studies. Generally, the estimates based on surface $CO_2$

measurements show carbon sinks or weak source, which are mainly in the range of -0.26 to 0.32 PgC yr$^{-1}$ (Valentitni et al., 2014; Jacobson et al., 2020), while the estimates from satellite $XCO_2$ retrievals report strong carbon sources, with values mainly in the range of 0.61 to 2.2 PgC yr$^{-1}$ (Liu et al., 2021; Palmer et al., 2019). Peiro et al. (2022) also found a similar phenomenon by comparing the carbon fluxes constrained using in-situ observations and OCO-2 retrievals within the same inversion frameworks. Although the estimates based on surface measurements are much closer to Ciais et al. (2021)'s result, the surface $CO_2$ observation sites in Africa are very sparse, there are only 4 stations over the African continent and 2 stations located in adjacent islands, indicating that the constraints from surface measurements are very poor, and the inverted fluxes often reflect the prior fluxes used in these inversions (Valentitni et al., 2014). In our prior flux, the NBE in Africa is 0.34 PgC yr$^{-1}$, that is consistent with above surface-based estimates. This indicates that the strong carbon source is almost constrained from satellite $XCO_2$. Since there is no TCCON site in Africa, which is usually used to verify and correct the satellite $XCO_2$ retrievals, leading larger uncertainties in the $XCO_2$ products, thus probably resulting in an overestimation of the surface flux. Peiro et al. (2022) reported that the version of OCO-2 retrievals had a significant effect on the inversion results in Africa. However, due to the lack of validation data for $XCO_2$ and few in situ $CO_2$ measurements, it is hard to know for sure which is more accurate. In South Asia, we show a very weak sink of -0.05±0.10 PgC yr$^{-1}$, while Ciais et al. (2021) presented a moderate sink of -0.25 PgC yr$^{-1}$. Based on bottom-up and top-down methods, there have been many studies on NBE in South Asia in the past. Overall, the bottom-up estimates are in the range of -0.01 ~ -0.25 PgC yr$^{-1}$ (Cervarich et al., 2016; Ciais et al., 2021; Nayak et al., 2015; Gahlot et al., 2017; Patra et al., 2013), while the top-down estimates are in the range of 0.04 ~ -0.37 PgC yr$^{-1}$ (Patra et al., 2013; Thompson et al., 2016; Cervarich et al., 2016; Niwa et al. 2012; Jiang et al., 2014; Swathi et al. 2021). Our result for South Asia is in the range of these previous studies.

### 5.3 Interannual variations and seasonal cycles

Figure 7a, b, and c show interannual variations (IAV) of the NEE in the NL, TL and SL, respectively. In NL, the IAV of NEE is relatively small, with an interannual amplitude of 1.09 ± 0.50 PgC/yr. The smallest year of NEE appeared in 2018, which was -1.87 ± 0.38 PgC/yr, and the largest year appeared in 2014, with value of -2.91 ± 0.33 PgC yr$^{-1}$. In TL, the inter-annual variability is very large, with the biggest NEE in 2011 of -2.27 ± 0.33 PgC yr$^{-1}$ and the smallest NEE in 2016 only -0.31 ± 0.41 PgC yr$^{-1}$. The interannual amplitude of NEE in TL is nearly twice that of NL, which reaches 1.96 ± 0.53 PgC yr$^{-1}$. The strongest carbon sink in 2011 and weakest sink in 2016 are related to the strongest 2011 La Niña and 2015/2016 El Niño events, respectively, which is in good agreement with many previous findings (Liu et al. 2017; Bastos et al. 2018; Wang et al., 2018; Koren et al., 2018). Bastos et al. (2018) showed a smaller difference of carbon fluxes between 2015 and 2011 using both bottom-up and top-down approaches, which was in the range of 0.7 ~ 1.9 PgC yr$^{-1}$. With the constraints of GOSAT and OCO-2 $XCO_2$, Liu et al. (2017) found that relative to the 2011 La Niña, the pantropical biosphere released 2.5 ± 0.34 PgC more carbon into the atmosphere in 2015, and during the peak 2015–2016 El Niño between May 2015 and April 2016, the more

released carbon reached 3.3 ± 0.34 PgC. In this dataset, the changes of carbon flux between 2011 La Niña and 2015-2016 El Niño events in the pantropical area are lower than the estimates of Liu et al. (2017), but close to Bastos et al. (2018). We estimate the change of NBE between 2015 and 2011 is $1.59 \pm 0.34$ PgC yr$^{-1}$, and the peak period of 2015-2016 El Niño released 2.79 PgC more than in 2011 (Figure S8). In addition, it also could be found there are weak carbon sinks in 2010 and 2019 in TL. There have been many studies on the decline of carbon sinks in tropical regions in 2010 (van der Laan-Luijkx et al., 2015; Doughty et al., 2015; Gatti et al., 2014). In 2019, the decrease of NEE may be related to the Indian Ocean Dipole event, which has significantly reduced the carbon uptakes over southern China, Indo-China peninsula, and Australia (Wang et al., 2021b). In SL, due to the small land area, its NEE is an order of magnitude lower than the other two regions. It could be found that there is a continuous decreasing trend. This trend is basically consistent with that in Australia (Figure 8j), indicating that the IAV of NEE in SL is dominated by that in southern Australia, especially in southeastern Australia (Byrne e t al., 2021). Previous studies have revealed that the enhanced carbon uptake in Australia from 2010 to 2012 was associated with the La Niña phase from the end of 2010 to early 2012 (Detmers et al., 2015), while the significantly increased carbon loss in 2019 was due to extreme drought (Byrne e t al., 2021) associated with the Indian Ocean Dipole event (Wang et al., 2021b), indicating that the decreasing trend of carbon sink in SL is caused by the extreme climate events occurred in the start and end years of this decade, respectively, thus this downtrend is just a coincidence. On average, the NEE in NL, TL, and SL during this decade are -2.33 ± 0.35, -1.25 ± 0.38, and -0.05 ± 0.07 PgC yr$^{-1}$, which account for 62.6%, 33.4% and 1.4% of the global total land sink, respectively, indicating that the global land NEE is dominated by the NEE in NL. However, the correlation coefficients between the IAVs of NEE in these three regions (NL, TL, and SL) and the IAV of global terrestrial NEE are 0.57, 0.86, and 0.37, respectively, indicating that the IAV of global NEE is dominated by its inter-annual changes in TL.

In Figure 8, we further present the IAVs and seasonal cycles of NEE in the 11 TRANSCOM regions. Since there are some overlaps between the TRANSCOM and RECCAP2 regions, for example, the N. America region in RECCAP2 is almost the sum of the boreal and temperate N. America, the Africa region in RECCAP2 is the sum of the northern and southern Africa in TRANSCOM. Besides, the IAVs of NEE in some regions of RECCAP2 like Russia, East Asia are dominated by the NEE changes in corresponding regions in TRANSCOM. Therefore, here we only analyze the annual and monthly changes of NEE in the TRANSCOM regions. The differences for the IAVs between the prior and posterior NEE in each region are shown in Figure S9.

There are significant differences in the IAVs of annual NEE in each region. For example, in boreal N. America, there is the weakest sink in 2016 and the strongest sink in 2017, while in temperate N. America, the weakest sink occurs in 2018, and the strongest in 2010; Europe has the weakest sink in 2018, but the strongest sink is in 2014. For the interannual amplitudes, temperate N. America, tropical S. America, southern Africa, Australia and Europe have relatively larger interannual amplitudes, with values above 0.6 PgC yr$^{-1}$; in temperate S. America, boreal Asia, northern Africa, temperate Asia and tropical Asia, the

interannual amplitudes are comparable, ranging from 0.33 to 0.40 PgC yr$^{-1}$, while in boreal N. America, it has the smallest interannual amplitude of 0.22 PgC yr$^{-1}$. Except for boreal N. America, boreal Asia, and Europe, the interannual amplitudes in other regions are larger than their ten-year averaged carbon sinks, especially in the temperate S. America, northern and southern Africa, and Australia, their inter-annual amplitudes of NEE reach more than 5 times of the mean carbon sinks.

For the seasonal cycles, the northern middle and high latitudinal regions have similar pattern, with carbon sources during the cold season (from October to April), and carbon sinks during the warm season (from May to September). In the cold season, the difference of carbon releases in different regions is relatively small, but in the warm season, the intensity of carbon sinks in different regions is significantly different, and the months in which the strongest carbon sinks appear are also different. Boreal Asia, temperate and boreal N. America have the strongest sinks in July, Europe has the strongest one in June, while temperate Asia has the strongest in August. For the southern lands, southern Africa and temperate S. America have a similar seasonal cycle. their carbon sources occur from July to about December, with peak in October, and carbon sinks appear from January to May. In Australia, the carbon sinks mainly occur from March to October. In tropics, northern Africa has an opposite seasonal cycle with its adjacent region of southern Africa, its carbon sink occurs during June to November. The seasonal cycles in tropical Asia and tropical S. America are also nearly opposite. Tropical S. America has the strongest sink in September and October, while tropical Asia has the strongest carbon release in October. In general, the tropical regions have a smaller seasonal amplitude, while the high latitudes have a larger seasonal amplitude. In boreal Asia and Europe, their seasonal amplitudes reach 1.17 and 0.96 PgC mo$^{-1}$, respectively, while in tropical Asia and tropical S. America, the seasonal amplitudes are only about 0.12 PgC mo$^{-1}$. The same region has basically similar seasonal cycles in different years, but the intensity of its carbon sources and sinks, the time of transition from carbon source to carbon sink, and the months with the strongest sink or source are also significantly different in different years. For example, in tropical Asia, the carbon sources from January to April in 2010 and 2016 are significantly stronger than those in normal years; in temperate N. America, the carbon sinks in the spring of 2012 are significantly stronger than normal, but the carbon sinks in the summer are significantly weaker than normal.

Generally, the IAVs of annual NEE and seasonal cycles are related to large-scale climate anomalies and regional extreme climate events like droughts, heatwaves and precipitation, which have been widely studied around the world (e.g., Ciais et al., 2005; Betts et al., 2020; Bastos et al., 2018; Koren et al., 2018; Reichstein et al., 2013; Frank et al., 2015; Zhao and Running, 2010). Evidences have shown that severe drought events occurred in Amazon in 2010 (Potter et al., 2011; Doughty et al., 2015), Europe in 2010 (Bastos et al., 2020a), 2012 (He et al., 2019) and 2018 (Bastos et al., 2020b; Graf et al., 2020; Wang et al., 2020), the United States in 2011-2012 (He et al., 2018; Wolf et al., 2016; Liu et al., 2018; Byrne et al., 2020) and 2018 (Li et al., 2020), and Australia in 2019 (Byrne et al., 2021) had caused significant reductions of terrestrial carbon uptakes. Accordingly, as shown in Figure 8, the NEE in this dataset are also much smaller in those years and regions compared with the normal year. Specially, in 2012, the contiguous United States experienced exceptionally warm temperatures and the most

severe drought since the Dust Bowl era of the 1930s, Wolf et al. (2016) found that the warm spring reduced the impact of the summer drought on net annual carbon uptake across the United States. As mentioned above, our dataset also shows the significant increase of carbon sink in the spring of 2012, and large decrease during the summertime in temperate N. America. In the summer 2010, western Russia was hit by an extraordinary heat wave, with the region experiencing by far the warmest July since records began (Otto, et al., 2012; Guerlet et al., 2013; Ishizawa et al., 2016), correspondingly, we find that in our dataset, the carbon sink in boreal Asia in July 2010 is the weakest in this decade, and the areas with significant positive anomaly of NEE (source increase) are mainly in western Russia (Figure S10). The strong El Niño events in 2015 and 2016 led to a significant reduction in carbon sinks in the pantropical regions, and many regions even turned to carbon sources (Liu et al., 2017; Bastos et al., 2018). Clearly, during 2015 – 2016, the inverted carbon sink in this study is much weaker than normal years in tropical S. America and tropical Asia, and it turns to carbon sources in northern and southern Africa. These indicate that this NEE dataset could clearly reveal the impact of climate extremes on carbon uptakes, thus it will benefit for the studies of the trends and drivers of carbon flux in different regions of the world.

## 6 Evaluations

### 6.1 Against surface flask observations

As shown in section 3, surface flask observations from 74 sites are used to evaluate the inversion results. The modeled $CO_2$ concentrations were extracted from the simulated 3-hour interval 3-D $CO_2$ fields according to the locations, time and heights of each observation. It should be noted that the records with absolute biases between the posterior $CO_2$ concentrations and $CO_2$ measurements greater than 10 ppm were removed, which are considered to be lack of regional representativeness. Due to the low spatial resolution (1.9°×2.5°) of our model, we cannot reproduce such observations. Figure 9 shows the comparisons between the posterior $CO_2$ concentrations and surface flask $CO_2$ measurements. At most sites located in ocean areas, tropical lands, and southern lands, the BIAS is within ±0.5 ppm, and MAE lower than 1 ppm. In the northern mid-high latitudes, BIAS of some stations is higher than ±1.0 ppm, and MAE of almost all stations is higher than 1.5 ppm (Table S1). The global mean BIAS, MAE, and RMSE are 0.36, 1.76, and 2.28 ppm. The CORR of each site are in the range of 0.86 and 1, with global mean of 0.96.

The higher deviations in the northern mid-high latitudes, especially in temperate N. America and Europe, are probably due to the mismatch of spatial and temporal representativeness between the observations and simulations. In order to further increase the spatial and temporal representativeness of the observations, regional and monthly mean observed and modeled concentrations in 7 land regions are compared. As shown in Figure 1, the 7 regions are high latitudes (> 60° N), N. America, S. America, Europe, East Asia, Africa, and Australia. There are 8 sites in the high latitudes, 19 sites in N. America, 9 sites in Europe, 5 sites in East Asia, 3 sites in S. America, 5 sites in Africa, and 4 sites in Australia (Figure 1, Table S1). Figure 10

shows the time series of the monthly observed and modelled $CO_2$ concentrations in the 7 regions. Besides, the Mauna Loa Observatory (MLO) in Hawaii is a global background site, the comparisons of monthly mean concentrations at MLO are also shown in Figure 10. Clearly, the modeled regional and monthly mean $CO_2$ concentrations agree well with the observations. The mean BIAS are in the range of 0.1 to 0.56 ppm, and MAE and RMSE are in the range of 0.42 ~ 1.46, and 0.52 ~ 1.73 ppm, respectively. In S. America, Africa, and Australia, the posterior $CO_2$ concentrations are very consistent with the observations, with BIAS only in the range of 0.1 ~ 0.24 ppm, and MAE about 0.5 ppm. Among these regions, the deviations in Europe and high latitude regions are relatively larger, with MAE greater 1.4 ppm and RMSE about 1.7 ppm. Significant positive biases mainly occur during the winter. This is understandable, because in the winter at high latitudes, satellite observations are very scarce, leading to very insufficient constraints on the winter carbon flux. This indicates that there may be an overestimation of carbon releases at high latitudes in winter. At MLO, the simulations also agree well with the observations, with BIAS, MAE, and RMSE of 0.2, 0.46, and 0.57 ppm, respectively. Figure S11 shows the time series of biases in the 7 regions and at the MLO site, for comparison, the biases of prior $CO_2$ concentrations are also shown in this figure. Clearly, the biases of the simulated $CO_2$ concentrations are significantly decreased relative to the prior. It also could be found that there is an upward trend in the biases of the posterior $CO_2$ concentrations in all regions except East Asia, as well as at the MLO site. On global average (74 sites), the annual mean biases increase from -0.36 ppm in 2010 to 0.75 ppm in 2019, with uptrend slope of 0.115 ppm yr$^{-1}$ (Figure S12). By multiplying by a factor of 2.124 PgC ppm$^{-1}$ (Ballantyne et al., 2012), this bias accumulation rate is equal to 0.244 PgC yr$^{-1}$, which is very consistent with the 10-year averaged bias in the inverted global AGR given in Section 5.1 (0.25 PgC yr$^{-1}$). This uptrend is a result of a residual trend in the inversions fit to the GOSAT data. We analyzed the timeseries of the global averaged monthly mean posterior $XCO_2$ and GOSAT $XCO_2$ concentrations, and found that the mismatches between the posterior $XCO_2$ fields and GOSAT data also have an upward trend from 2010 to 2019, with an annual mean increment about 0.09 ppm yr$^{-1}$ (Figure S13).

**6.2 Against aircraft measurements**

We further evaluate the posterior $CO_2$ concentrations against the aircraft observations. First, the posterior $CO_2$ were extracted from the simulated $CO_2$ fields according to the locations, time and heights of each aircraft observation, and then, both the observed and modeled $CO_2$ concentrations were divided into 14 layers: 1000–1500, 1500–2000, 2000–2500, 2500–3000, 3000–3500, 3500–4000, 4000–4500, 4500–5000, 5000–5500, 5500–6000, 6000–7000, 7000–8000, 8000–9000 and above 9000 m (CONTRAIL only 3-10 layer, and CARBAM only 1-8 layer). Monthly mean observed and modeled $CO_2$ concentrations at each height were calculated and compared for the CONTRAIL and CARBAM profiles. For comparisons against the HIPPO observations, the data were further divided into 2° interval along longitudinal direction, and all data in each layer and 2° of latitudes were averaged.

Figure 11 and 12 shows the evaluation results of monthly mean profiles in the 8 cities over the Asia-Pacific region, and

at the 4 sites in the Amazon basin, respectively. Overall, the deviations between the simulations and observations decrease with height. In the Asia-Pacific region, the BIAS are basically within ±0.5 ppm, and most MAE are smaller than 1 ppm, especially in Southeast Asia, indicating that we have a good estimate of NEE in this area. Shanghai and New Delhi have relatively larger MAE and RMSE, with MAE about 1.5 ppm, and RMSE existing 2 ppm in the lowest level, probably due to the fact that Shanghai and New Delhi are one of the largest cities in China and India, respectively, and have very strong anthropogenic $CO_2$ emissions, which may affect the performance of the MOZART model. In the Amazon basin, the MAE and RMAE of all 4 sites decrease with height, with MAE and RMSE decreasing from about 2 ppm near 1000 m height to about 1.5 ppm near 4000 m. For BIAS, below 2000 m, they increase significantly with height. There are negative (~ -1.0 ppm, data not shown), small (~ 0.2 ppm), and significant positive BIAS (~ 0.9 ppm) below 1000 km, at 1000 ~ 1500 m, and 1500 ~ 2000 m heights, respectively, indicating that there are considerable vertical transport errors, and the carbon sinks over tropical S. America may have systematic biases.

Figure 13 shows the comparisons against the HIPPO observations at different heights and latitudes. Overall, most BIAS are within ±0.5 ppm, showing a good agreement between the simulations and observations. Relatively large BIAS occurs over northern high latitudes, which is consistent with the comparisons against the surface observations as shown in Figure 10, and also reveals an overestimation of carbon releases at high latitudes.

**7 Summary**

A global NEE dataset is essential for estimating the regional terrestrial carbon budget and understanding the responses of carbon fluxes to extreme climates. Here, by assimilating the GOSAT ACOS v9 $XCO_2$ product, we generate a ten-year global monthly terrestrial NEE dataset from 2010 to 2019 (GCAS2021) using the GCASv2 system. GCAS2021 includes monthly and annual gridded (1°×1°) prior and posterior NEE and OCN flux, and prescribed FIRE and FFC emissions, and globally, latitudinally, and regionally aggregated fluxes and their uncertainties. Globally, the decadal mean NBE and AGR as well as their IAVs match well with the estimates of GCP2020. Regionally, our product shows carbon sinks over eastern N. America, Amazon, Congo Basin, Europe, boreal forests, southern China, and southeast Asia, and carbon sources over western US, African grasslands, Brazilian plateau, and parts of South Asia. In the 11 TRANSCOM land regions, the NBEs of temperate N. America, northern Africa and boreal Asia are between the results of CMS-Flux NBE 2020 and CT2019B, and those in temperate Asia, Europe, and tropical Asia are very close to CMS-Flux NBE 2020 but significantly different from CT2019B. In the RECCAP2 regions, except for Africa and South Asia, the NBEs are comparable with the latest bottom-up estimate of Ciais et al. (2021). The IAVs and seasonal cycles of NEE could clearly reflect the impact of extreme climates or large-scale climate anomalies. We also qualitatively evaluate the NEE estimates by comparing posterior $CO_2$ concentrations with independent $CO_2$ measurements from surface flask and aircraft $CO_2$ observations, and the results show that the simulated

remote site and regional average $CO_2$ concentrations, as well as the vertical $CO_2$ profiles, are all consistent with the observations. We believe that this dataset will be useful in the estimates of regional or national-scale terrestrial carbon budgets, the study of carbon sink evolution mechanisms, the evaluation of ecosystem models, and the assessments of carbon neutrality strategies.

**Data availability**


The GCAS2021 data are available at https://doi.org/10.5281/zenodo.5829774 (Jiang, 2022). The regional aggregated fluxes are provided as xlsx files with file size ~135 KB, the gridded fluxes and ensemble members are provided in NetCDF format with file size ~82 MB and 5.8 GB, respectively.

**Author contributions**


FJ, JC and WJ designed the research; FJ run the model, analyzed the results and wrote the paper; HW handled the GOSAT $XCO_2$ retrievals; WH analyzed the products of CMS-Flux and CT2019B; WJ run the BEPS model; MJ, SF, and LZ participated in evaluations; JC, WJ, MW and HW participated in the discussions of the inversion results and provided inputs on the paper for revision before submission.

**Competing interests**


The authors declare that they have no conflict of interest.

**Acknowledgements**

This work is supported by the National Key R&D Program of China (Grant No: 2020YFA0607504 and 2016YFA0600204), and the Fundamental Research Funds for the Central Universities (Grant No: 090414380030, 090414380027, and 0207-14380179). We acknowledge all atmospheric data providers to obspack_co2_1_GLOBALVIEWplus_v6.0_2020-09-11.


CarbonTracker CT2019B results are provided by NOAA ESRL, Boulder, Colorado, USA, from the website at http://carbontracker.noaa.gov. The GOSAT data are produced by the OCO project at the Jet Propulsion Laboratory, California Institute of Technology, and obtained from the data archive at the NASA Goddard Earth Science Data and Information Services Center. We are also grateful to the High-Performance Computing Center (HPCC) of Nanjing University for doing the numerical calculations in this paper on its blade cluster system.

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

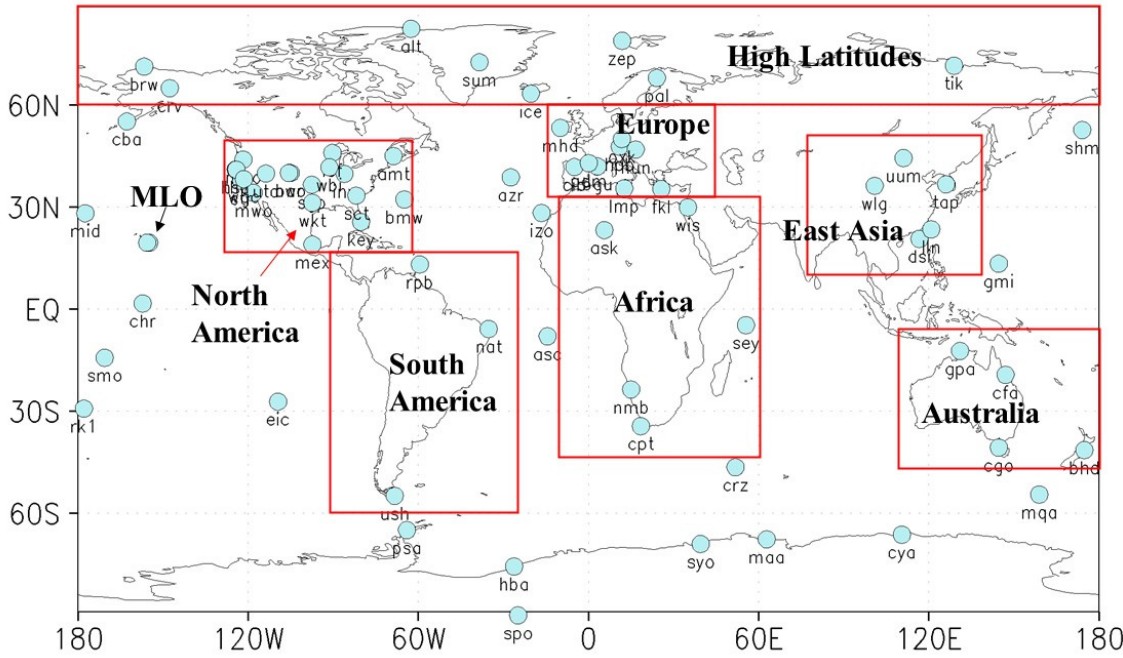

**Figure 1: Distributions of the surface flask observation sites used in this study.**

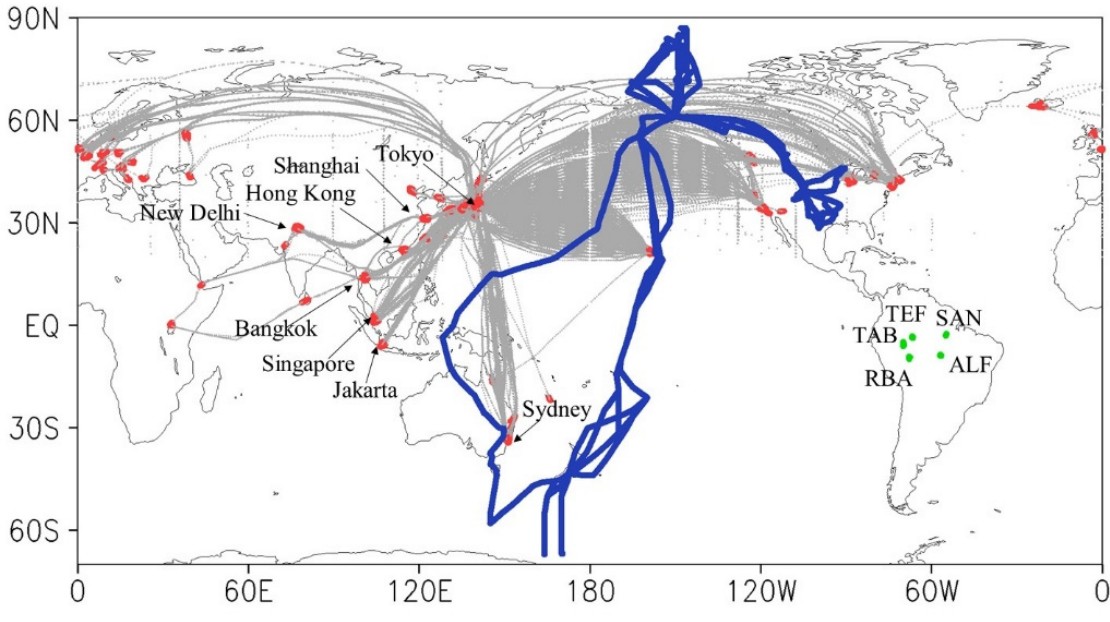

**Figure 2**:**Locations of aircraft observations (red and gray, observations of the CONTRAIL project, in while red marks show observations below 6 km; dark blue, observations of the HIPPO project; green, data of the CARBAM project).**

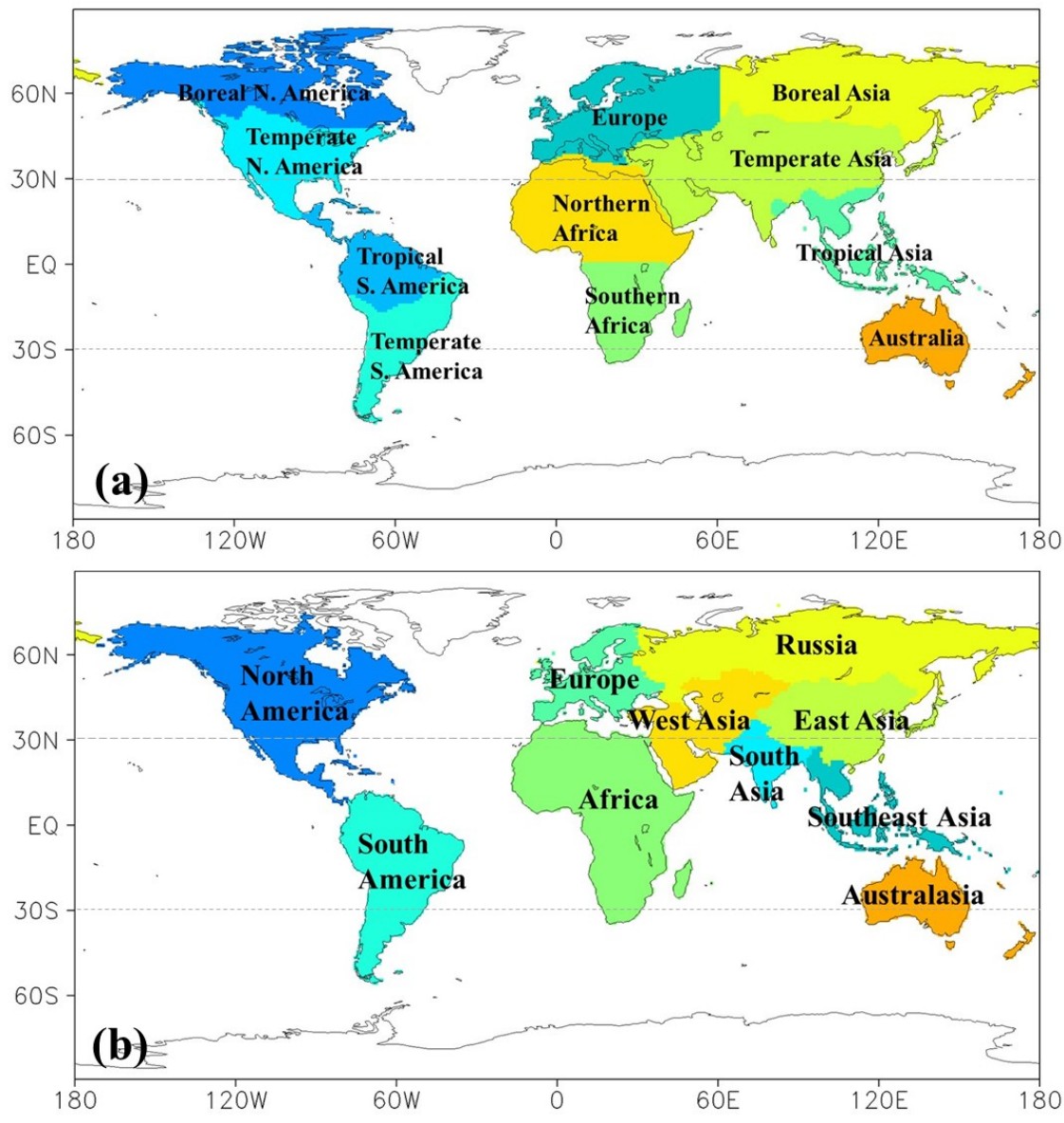

**Figure 3: Map of regional masks used in calculating regional fluxes, (a) the TRANSCOM region, (b) the RECCAP2 region.**


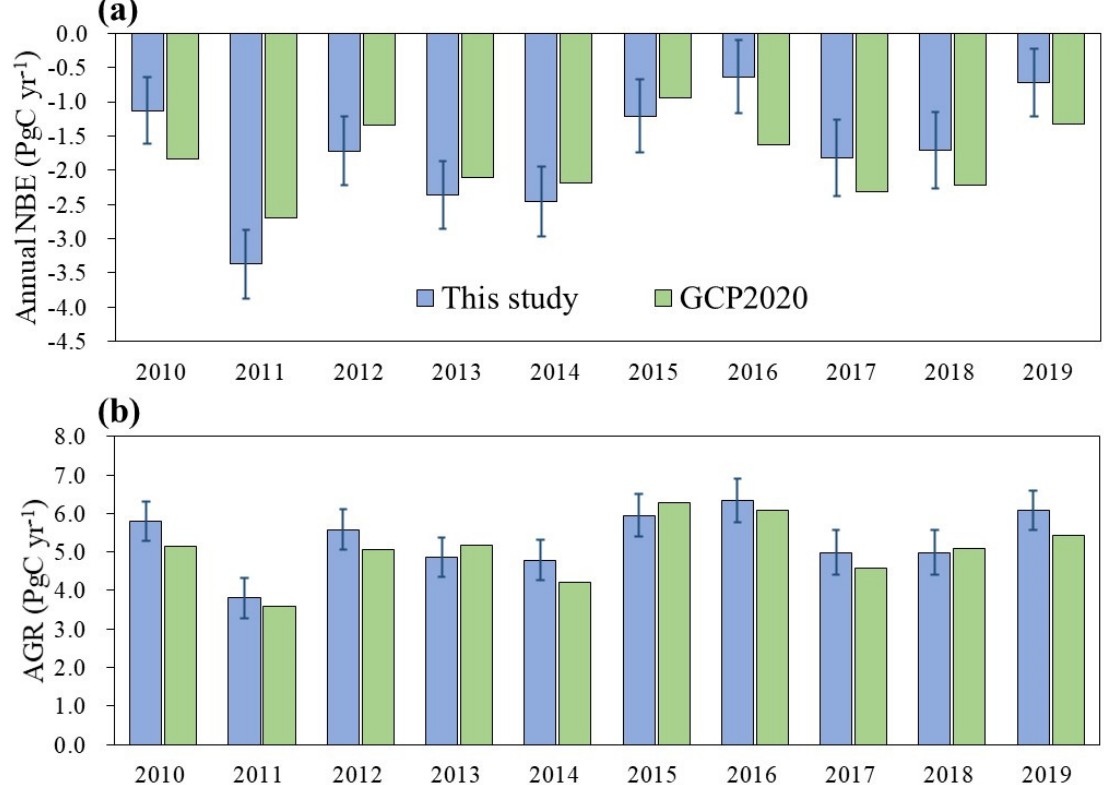

**Figure 4: Comparisons between this study and GCP2020 for (a) NBE and (b) Atmospheric Growth Rate (AGR), the NBE of GCP2020 is the sum of land sink and land-use change carbon emission, the AGR of this study is the Net Flux as listed in Table 1.**

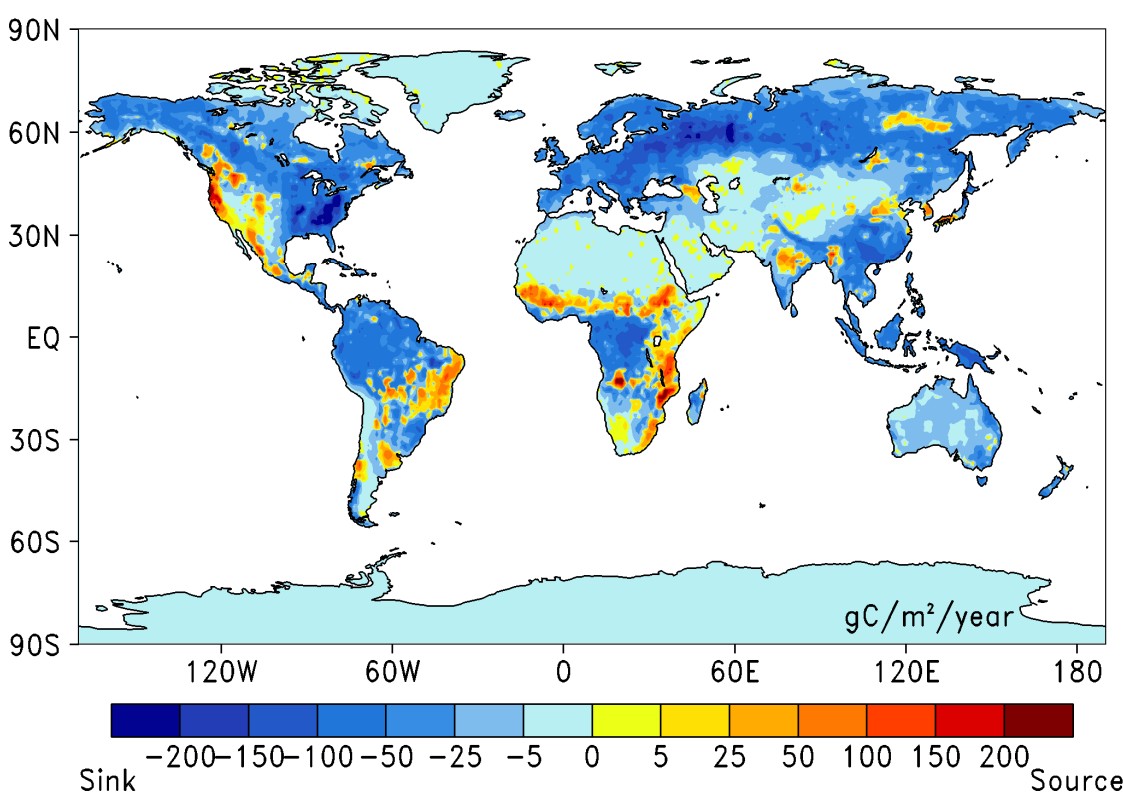


**Figure 5: Global distribution of mean annual NEE during 2010-2019.**

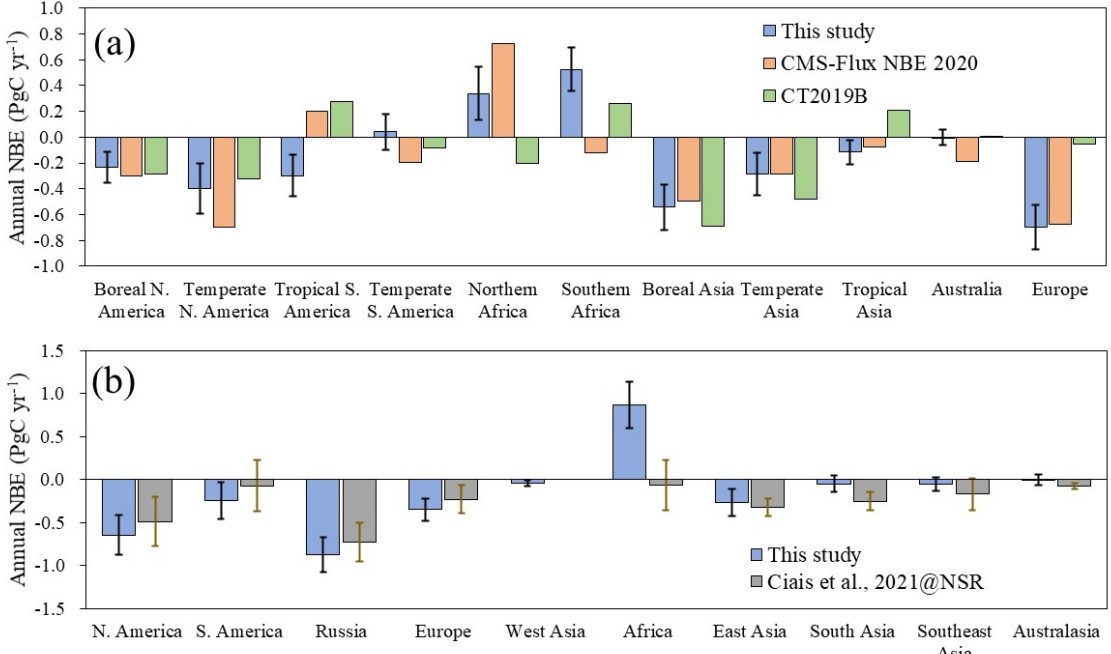

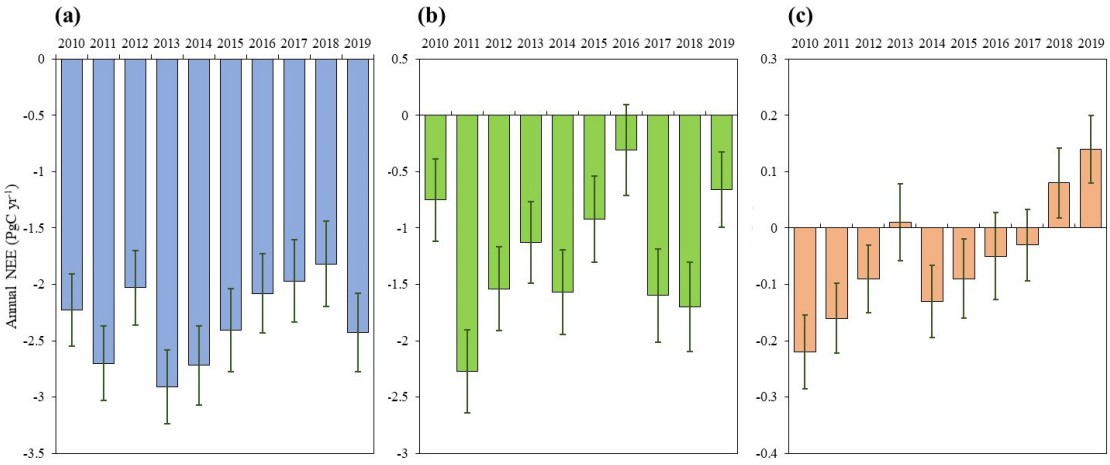

**Figure 6: 2010-2019 averaged regional NBE in a) the TRANSCOM regions, and b) the RECCAP2 regions, both CMS-Flux NBE 2020 and CT2019B are averaged from 2010 to 2018, the result of Ciais et al. (2021) is a bottom-up estimate, which is for the period of 2000-2009.**

**Figure 7: Interannual variations of annual NEE of different latitudes (a, northern mid-high latitudes (> 30° N); b, tropical latitudes (30° S ~ 30° N); c, southern middle latitudes (<30° S)).**

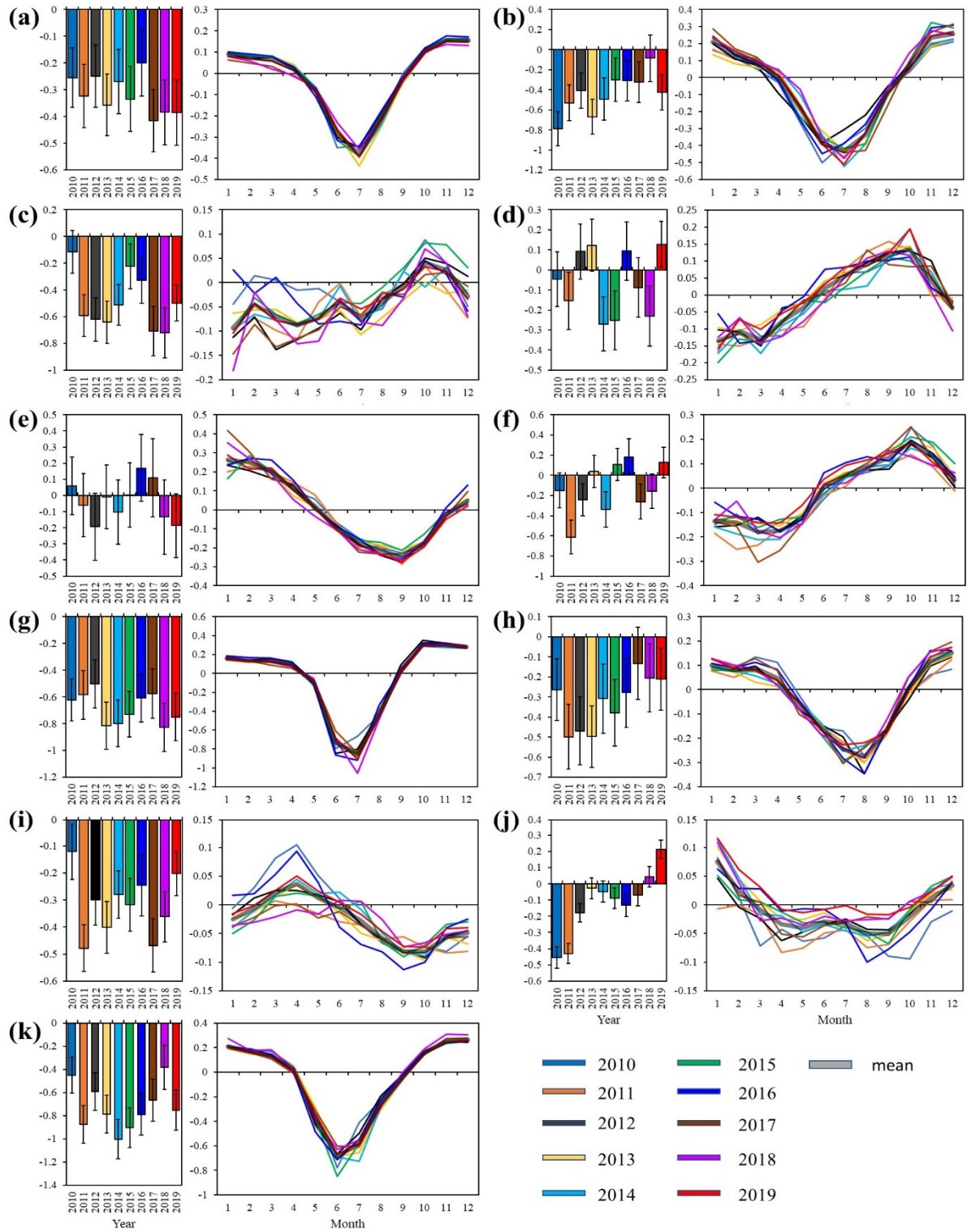


**Figure 8: Interannual variations of the annual (unit: PgC yr$^{-1}$) and monthly (unit: PgC month$^{-1}$) NEE in the 11 TRANSCOM regions (a, boreal N. America; b, temperate N. America; c, tropical S. America; d, temperate S. America; e, northern Africa; f, southern Africa; g, boreal Asia; h, temperate Asia; i, tropical Asia; j, Australia; k, Europe).**

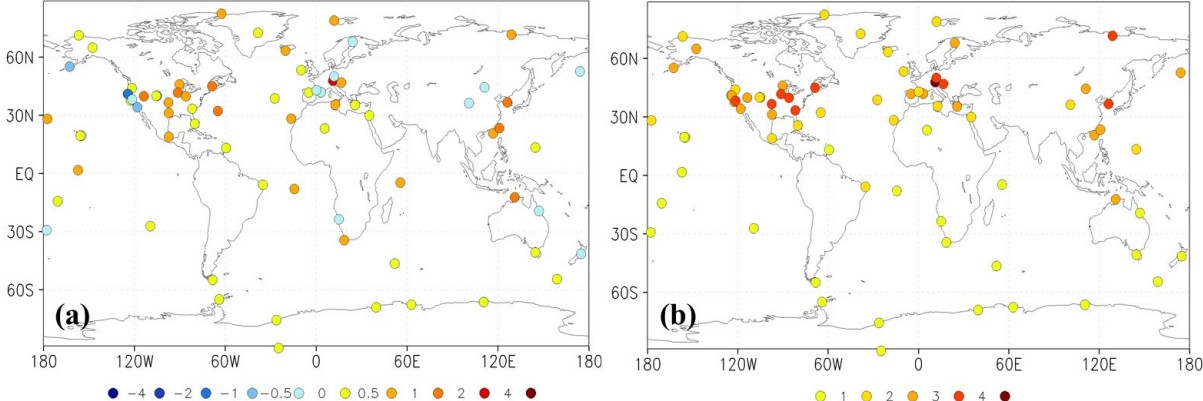

Figure 9: Spatial distributions of the (a) BIAS and (b) MAE of the posterior $CO_2$ concentrations at each site (simulations minus observations, unit: ppm)

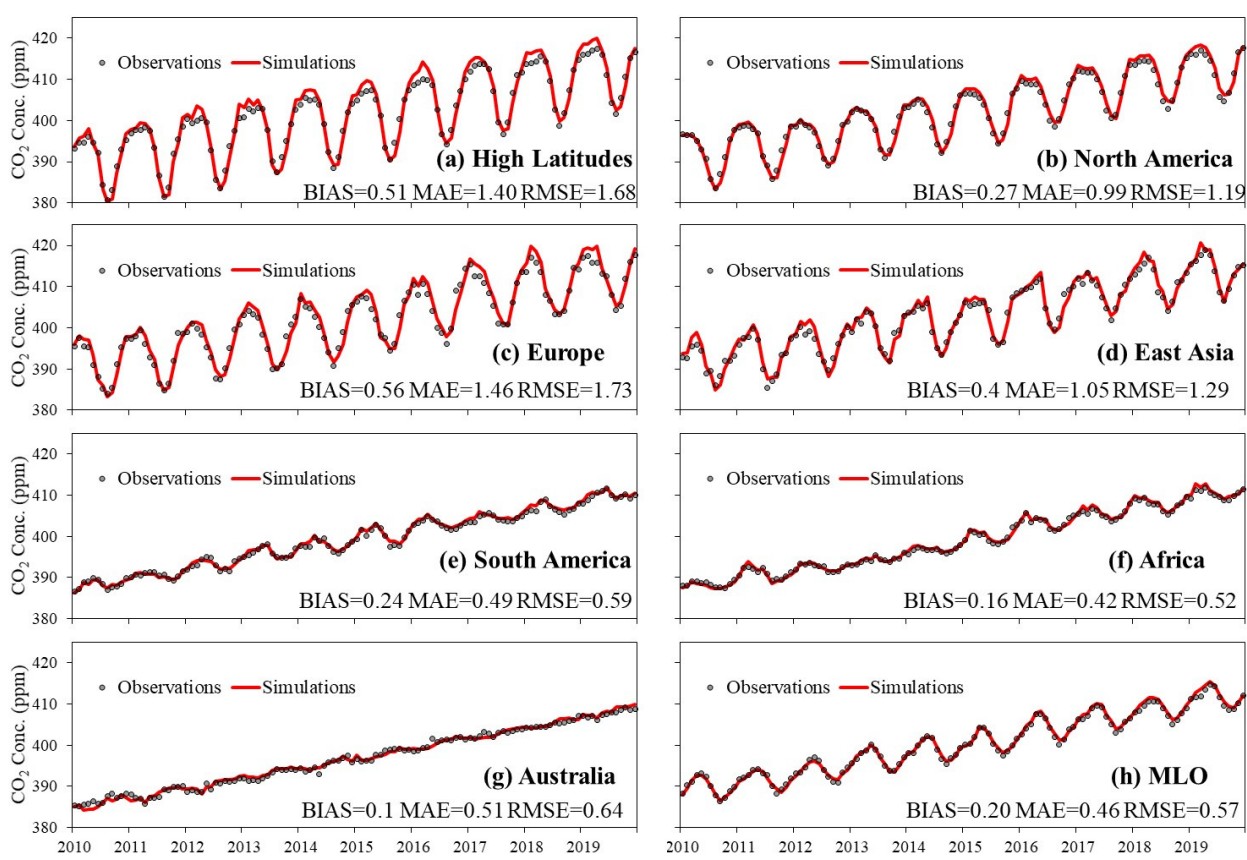

Figure 10: Time series of monthly averaged observations and simulations in the 7 regions and MLO site.

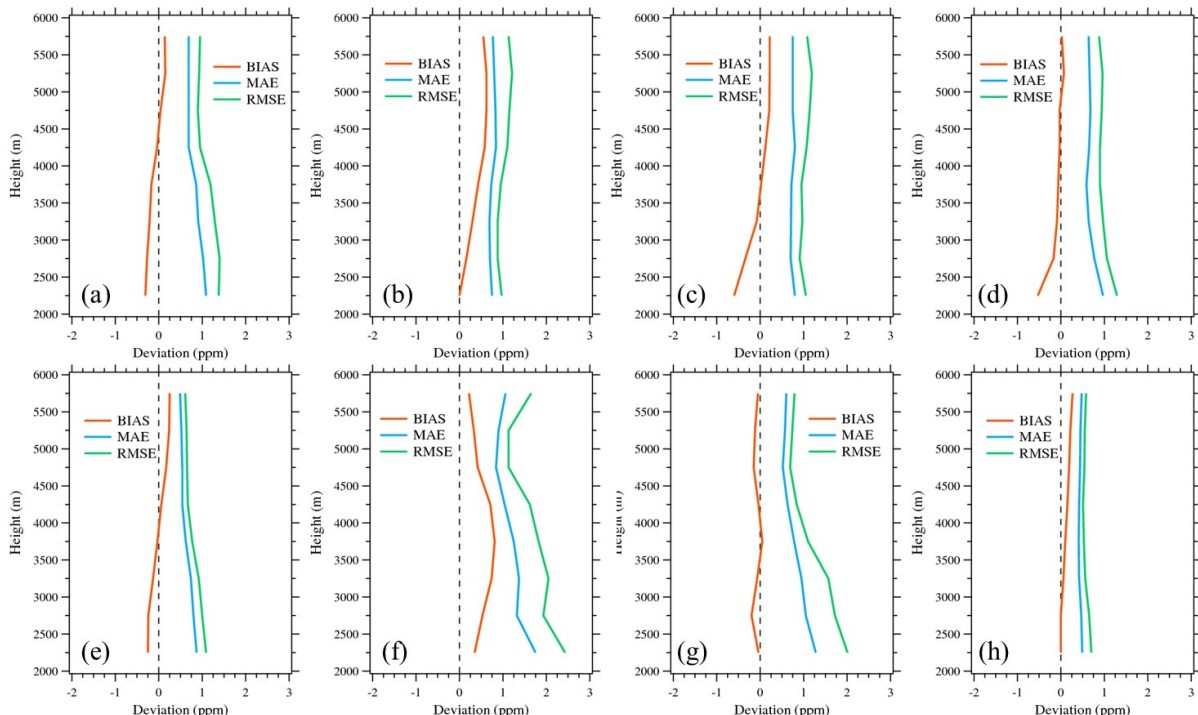


**Figure 11:Statistical results for monthly mean profiles in the 8 Asia-Pacific cities (a, Hong Kong; b, Bangkok; c, Singapore; d, Jakarta; d, Tokyo; f, Shanghai; g, New Delhi; h, Sydney).**

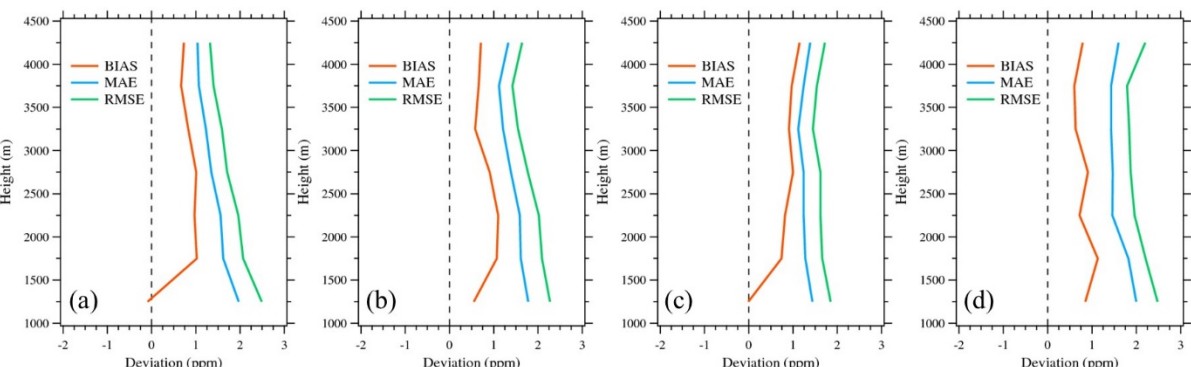

**Figure 12: Statistical results at different heights against the observations in the Amazon basin (a, ALF; b, RBA; c, SAN; d, TAB_TEF).**

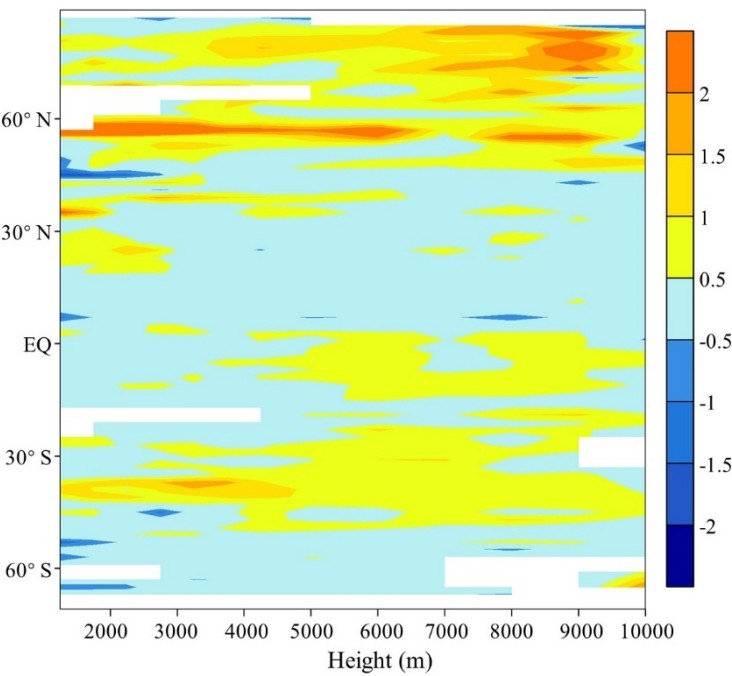

**Figure 13: BIAS at different latitudes and heights against the HIPPO observations.**







**Table 1: Global carbon budget (PgC yr⁻¹).**

| Year | NEE | OCN flux | FFC emission | FIRE emission | NBE | Net Flux | NBE | |
|---|---|---|---|---|---|---|---|---|
| | | | | | | | CMS-Flux NBE 2020 | CT2019B |
| 2010 | -3.28±0.49 | -2.11±0.15 | 9.04 | 2.15 | -1.13±0.49 | 5.80±0.51 | -2.1 | -0.9 |
| 2011 | -5.24±0.50 | -2.21±0.15 | 9.40 | 1.87 | -3.37±0.50 | 3.81±0.52 | -3.71 | -2.56 |
| 2012 | -3.77±0.50 | -2.27±0.15 | 9.58 | 2.05 | -1.72±0.50 | 5.58±0.52 | -2.05 | -0.89 |
| 2013 | -4.13±0.49 | -2.40±0.15 | 9.63 | 1.77 | -2.36±0.49 | 4.86±0.51 | -2.13 | -1.89 |
| 2014 | -4.50±0.51 | -2.46±0.16 | 9.71 | 2.04 | -2.46±0.51 | 4.79±0.53 | -3.82 | -2.39 |
| 2015 | -3.50±0.53 | -2.52±0.16 | 9.68 | 2.29 | -1.21±0.53 | 5.95±0.56 | -0.56 | -0.84 |
| 2016 | -2.51±0.53 | -2.73±0.16 | 9.71 | 1.87 | -0.64±0.53 | 6.33±0.56 | -0.85 | -0.27 |
| 2017 | -3.74±0.56 | -3.06±0.17 | 9.87 | 1.92 | -1.82±0.56 | 4.99±0.58 | -2.05 | -1.41 |
| 2018 | -3.54±0.56 | -3.37±0.16 | 10.07 | 1.83 | -1.71±0.56 | 4.99±0.58 | -1.77 | -1.86 |
| 2019 | -3.04±0.49 | -3.23±0.16 | 10.03 | 2.32 | -0.72±0.49 | 6.08±0.52 | - | - |
| Mean | -3.73±0.52 | -2.64±0.16 | 9.67 | 2.01 | -1.71±0.52 | 5.32±0.54 | -2.12 | -1.45 |


**Table 2: Regional terrestrial ecosystem carbon flux (PgC yr⁻¹).**

| TRANSCOM region | NEE | FIRE | NBE | RECCAP2 region | NEE | FIRE | NBE |
|---|---|---|---|---|---|---|---|
| Boreal N. America | -0.32±0.12 | 0.08 | -0.23±0.12 | N. America | -0.78±0.23 | 0.14 | -0.64±0.23 |
| Temperate N. America | -0.43±0.19 | 0.04 | -0.40±0.19 | S. America | -0.53±0.21 | 0.29 | -0.24±0.22 |
| Tropical S. America | -0.50±0.16 | 0.20 | -0.30±0.16 | Russia | -1.02±0.20 | 0.15 | -0.87±0.20 |
| Temperate S. America | -0.06±0.14 | 0.10 | 0.04±0.14 | Europe* | -0.36±0.13 | 0.01 | -0.35±0.13 |
| Northern Africa | -0.03±0.21 | 0.37 | 0.34±0.21 | West Asia | -0.05±0.03 | 0.01 | -0.04±0.03 |
| Southern Africa | -0.13±0.17 | 0.66 | 0.53±0.17 | Africa | -0.17±0.27 | 1.03 | 0.87±0.27 |
| Boreal Asia | -0.68±0.18 | 0.14 | -0.54±0.18 | East Asia | -0.30±0.15 | 0.03 | -0.27±0.15 |
| Temperate Asia | -0.32±0.17 | 0.04 | -0.29±0.17 | South Asia | -0.07±0.10 | 0.02 | -0.05±0.10 |
| Tropical Asia | -0.32±0.09 | 0.20 | -0.12±0.09 | Southeast Asia | -0.25±0.08 | 0.19 | -0.06±0.08 |
| Australia | -0.12±0.06 | 0.12 | 0.00±0.06 | Australasia | -0.12±0.06 | 0.12 | 0.00±0.06 |
| Europe | -0.72±0.17 | 0.02 | -0.70±0.17 | - | - | - | - |

*Excluding European Russia