# Peer review of "A ten-year global monthly averaged terrestrial NEE inferred from the ACOS GOSAT v9 XCO2 retrievals (GCAS2021)"

_Earth System Science Data, 2022_

## Author Comment (AC1)

**Referee #1**

We would like to thank the anonymous referee for his/her comprehensive review and valuable suggestions. These suggestions help us to present our results more clearly. In response, we have made changes according to the referee's suggestions and replied to all comments point by point. All the page and line number for corrections are referred to the revised manuscript with tracked changes, while the page and line number from original reviews are kept intact.

This study presents an NEE dataset based on flux inversion analysis that assimilate 10 years of ACOS GOSAT v9 XCO2 retrievals. To my knowledge, this is the first published flux dataset using the full ACOS GOSAT v9 XCO2 retrievals and believe it to be a signficiant contribution to the community. I find that the dataset is well described and evaluated, and recommend publication after addressing several generally minor comments.

My most significant comment is that the atmospheric growth rate reported here seems to deviate substantially from that reported by GCP2020. From line 227 it is stated that the bias (GCAS2021 minus GCP2020) is 0.25 PgC/yr. Over the 10 year inversion, this amounts to 2.5 PgC or 1.2 ppm (using 1 ppm / 2.086 PgC from https://acp.copernicus.org/preprints/12/C8465/2012/acpd-12-C8465-2012.pdf). I would expect such a difference to be evident in the comparison against independent CO2 data (which does not appear to show this bias). It would be useful to also compare the growth rate with that reported by NOAA (https://gml.noaa.gov/ccgg/trends/gr.html) and discuss this difference some more. For the comparison against independent $CO_2$ data, it would be useful to show whether these mismatches are decreased relative to the prior (would be fine to have this in the supplement).

Response: Many thanks for this suggestion. We have checked the AGR of GCP2020, which was estimated directly from atmospheric $CO_2$ concentration measurements, and provided by the US NOAA Earth System Research Laboratory (NOAA/ESRL, http://www.esrl.noaa.gov/gmd/ccgg/trends/global.html) (Friedlingstein et al., 2020). Therefore, the growth rates reported by NOAA and GCP2020 are exactly the same. The growth rate reported in GCP2020 is in units of GtC $yr^{-1}$, which was converted from the report of NOAA in units of ppm $yr^{-1}$ by multiplying by a factor of 2.124 GtC $ppm^{-1}$. In this study, the mean bias of AGR between GCAS2021 and GCP2020 during the 10 years is 0.25 PgC/yr, accordingly, over the 10 years inversion, the accumulated bias in atmospheric $CO_2$ concentration should be 1.18 ppm (converted using 2.124 GtC $ppm^{-1}$). Such an accumulated bias could be found in the comparison against independent $CO_2$ data (Figure 10), however, since we only show the time series of monthly averaged observations and simulations, this error accumulation problem is not clearly presented in that figure. In the revised manuscript, we have added the time series of the biases

between observations and simulations in the revised supplement (Figure S11), which is also given in this text as Figure R1. Clearly, there is an upward trend in the biases between simulations and observations in all regions except East Asia, as well as at the MLO site. For a clearer presentation and to compare with the AGR bias, the inter-annual variations of the global averaged (74 surface flask $CO_2$ measurements selected for independent evaluation in this study, Section 3 and Figure 1) annual mean bias is showed in Figure R2. There is negative bias in 2010, with value of -0.36 ppm, and significant positive bias in 2019, with value of 0.75 ppm, that is, the bias increases by 1.1 ppm from 2010 to 2019. The slope of this uptrend is 0.115 ppm/yr, indicate that over the 10 year inversion, the accumulated bias reaches 1.15 ppm (0.115 ppm/yr $\times$ 10 yr). Both the biases calculated based on the difference between 2010 and 2019 (1.1 ppm) and estimated from uptrend slope (1.15 ppm) are very close with the one estimated from the mean bias of AGR between GCAS2021 and GCP2020 (i.e., 1.18 ppm). Figure R2 has also been added in the revised supplement, and named as Figure S12.

In Figure S11 (Figure R1), we also show the biases between observations and prior simulations. Clearly, the mismatches are significantly decreased relative to the prior. In the 7 regions, for the prior simulations, the biases and RMSE are in the range of 1.88~3.35 and 2.13~4.07 ppm, respectively, after constrained using $XCO_2$ retrievals, they are decreased to 0.1~0.56 and 0.52 ~ 1.68 ppm, respectively.

The following sentences are added in the revised manuscript.

In section 5.1 (see page 9, lines 248-251):

"… Additionally, GCP2020 also reported atmospheric growth rate (AGR) of $CO_2$ in the atmosphere, which was estimated directly from atmospheric $CO_2$ concentration measurements provided by the NOAA Earth System Research Laboratory (Friedlingstein et al., 2020). Ideally, the inverted global net carbon flux (i.e., AGR) should agree with the observed AGR. As shown in Figure 4,"

In section 6.1 (see page 16, lines 471-478):

"…Figure S11 shows the time series of biases in the 7 regions and at the MLO site, for comparison, the biases of prior $CO_2$ concentrations are also shown in this figure. Clearly, the biases of the simulated $CO_2$ concentrations are significantly decreased relative to the prior. It also could be found that there is an upward trend in the biases of the posterior $CO_2$ concentrations in all regions except East Asia, as well as at the MLO site. On global average (74 sites), the annual mean biases increase from -0.36 ppm in 2010 to 0.75 ppm in 2019, with uptrend slope of 0.115 ppm yr$^{-1}$ (Figure S12). By multiplying by a factor of 2.124 PgC ppm$^{-1}$ (Ballantyne et al., 2012), this bias accumulation rate is equal to 0.244 PgC yr$^{-1}$, which is very consistent with the 10-year averaged bias in the inverted global AGR given in Section 5.1 (0.25 PgC yr$^{-1}$)."

[Figure]

Figure R1. Time series of monthly averaged biases between observations and simulations and the frequency distribution of the biases in the 7 regions and MLO site (the black dotted line represents the linear trend of the biases between the observations and the posterior simulations)

[Figure]

Figure R2. Inter-annual variations of the global averaged annual mean bias (error bar represents standard deviation of monthly mean biases in one year; the dotted line is its linear trend)

L35: remove "could" in "ecosystems could uptake"

Response: we have removed "could" in the revised manuscript, see page 2, line 35.

L40: remove "," in NBE, = NEE + wildfire carbon emission

Response: we have removed "," in the revised manuscript, see page 2, line 40.

L77-78: Please double check wether Liu et al. (2021) actually optimizes fire emissions. I think they just optimize NEE but report NBE because errors in fire would alias into NEE.

Response: Thank you! We have checked the method of Liu et al. (2021). Their prior $CO_2$ fluxes include NBE, air–sea carbon exchange, and fossil fuel emissions, and only fossil fuel emissions were prescribed (i.e., assuming no uncertainty). The prior NBE was constructed using the CARDAMOM framework, which included fire emissions. In both their inversion framework and data product, the data of fire emissions are not independently presented.

L128: What is the data assimilation window length?

Response: The length of DA window is 1 week. We have added this information in page 5, line 140 in the revised manuscript.

L141: I did not see the qauntity "BIO" or "FCC" explicitly defined. Please make sure that these and other abbreviations are explicitly defined in the text.

Response: Thank you! We have changed "BIO" and "FCC" to "NEE" and "FFC" in that sentence in the revised manuscript (see page 5, lines 148-149). We also checked the full text to make sure all abbreviations are explicitly defined.

L162: "pfp" should be "PFP"

Response: Thank you. We have changed "pfp" to "PFP" in the revised manuscript, see page 6, line 172.

L210: "low latitudes (30 S ~ 30 N, TL)" should be "tropical latitudes (30 S ~ 30 N, TL),"

Response: Thank you! We have changed "low" to "tropical" in the revised manuscript, see page 8, line 220.

L238: "NEE constraint with" should be "NEE constrained with"

Response: Thank you! We have changed "constraint" to "constrained" in the revised manuscript, see page 10, line 269.

L254-261: "strongest NEE" should be "strongest sink". Similarly, "weakest NEE" should be "weakest sink"

Response: Thank you! We have changed "strongest" and "weakest" NEE to "strongest" and "weakest" sink in the revised manuscript, see page 10, lines 289-290.

L287-289: In general, OCO-2 $XCO_2$ or GOSAT $XCO_2$ flux inversions find northern sub-saharan Africa to be a strong source of $CO_2$, while in situ $CO_2$ inversion do not (e.g., https://doi.org/10.5194/acp-22-1097-2022). However, due to the lack of validation data for $XCO_2$ and few in situ $CO_2$ measurements, it is hard to know for sure which is more accurate.

Response: Thank you for this suggestion. There are big differences in top-down estimates of African NBE in different studies. Generally, the estimates based on surface in-situ measurements show carbon sinks or weak source, while the estimates from satellite $XCO_2$ retrievals report strong carbon sources. We strongly agree that due to the lack of validation data for $XCO_2$ and few in situ $CO_2$ measurements, it is hard to know for sure which is more accurate. We have added the following sentences in the revised manuscript (see page 12, lines 340-342, and lines 349-351).

"... Peiro et al. (2021) also found a similar phenomenon by comparing the carbon fluxes constrained using in-situ observations and OCO-2 retrievals within the same inversion frameworks. Although the estimates based on surface measurements"

"… thus probably resulting in an overestimation of the surface flux. Peiro et al. (2021) found that the version of OCO-2 retrievals also had a significant effect on the inversion results in Africa. However, due to the lack of validation data for $XCO_2$ and few in situ $CO_2$ measurements, it is hard to know for sure which is more accurate."

L324-325: I would be careful about calling this a "trend". It appears that this is primarily coming from Australia (Fig. 8j) which had very wet seasons at the start of the record (during 2011 La Nina) and has had drought for the last few years. So the "trend" might be strongly impacted by these events at the end points. Interestingly, the variability for Australia in this study looks very similar to the variability for southeast Australia reported in Fig 5 of https://doi.org/10.1029/2021AV000469.

Response: Many thanks for this comment. We strongly agree with you that this decreasing trend is primarily coming from Australia (Fig. 8j), which had very wet seasons at the start of the record (during 2011 La Nina) and had drought during the last few years. We also compared the interannual variations of NEE in Australia and in southeast Australia in our dataset, their interannual variations are indeed consistent. Southeastern Australia's carbon sink accounts for about 40% of Australia's carbon sink.

We have added the following sentences in the revised manuscript (see page 13, lines 377-383).

"… It could be found that there is a continuous decreasing trend. This trend is basically consistent with that in Australia (Figure 8j), indicating that the IAV of NEE in SL is dominated by that in southern Australia, especially in southeastern Australia (Byrne e t al., 2021). Previous studies have revealed that the enhanced carbon uptake in Australia from 2010 to 2012 was associated with the La Niña phase from the end of 2010 to early 2012 (Detmers et al., 2015), while the significantly increased carbon loss in 2019 was due to extreme drought (Byrne e t al., 2021) associated with the Indian Ocean Dipole event (Wang et al., 2021b), indicating that the decreasing trend of carbon sink in SL was caused by the extreme climate events occurred in the start and end years of this decade, respectively, thus this downtrend is just a coincidence."

L326: "global land NEE" should be "global land sink"

Response: Thank you! We have changed "global land NEE" to "global land sink" in the revised manuscript, see page 13, line 384.

L326-237: It is unclear which regions are being correlated for each of these numbers. Please state explicitly.

Response: We have revised that sentence to "The correlation coefficients between the IAVs of NEE in these three regions (NL, TL, and SL) and the IAV of global terrestrial NEE are 0.57, 0.86 and 0.37, respectively, indicating …" in the revised manuscript, see page 13, lines 385-386.

L368: "Unite States" should be "United States"

Response: We have changed "Unite States" to "United States" in the revised manuscript, see page 15, line 427.

L368: "Unite States in 2011-2012 (He et al., 2018; Wolf et al., 2016)". There are several more studies that used atmospheric $CO_2$ to study this event that could be cited:

Liu, J., Bowman, K., Parazoo, N. C., Bloom, A. A., Wunch, D., Jiang, Z., Gurney, K. R., & Schimel, D. (2018). Detecting drought impact on terrestrial biosphere carbon fluxes over contiguous US with satellite observations. Environmental Research Letters, 13(9), 095003.

Byrne, B., Liu, J., Bloom, A. A., Bowman, K. W., Butterfield, Z., Joiner, J., et al. (2020). Contrasting regional carbon cycle responses to seasonal climate anomalies

across the east-west divide of temperate North America. Global Biogeochemical Cycles, 34, e2020GB006598. https://doi.org/10.1029/2020GB006598

Response: Thank you! We have added these two citations in the revised manuscript, see page 15, line 427, and page 20, lines 578-580.

L375-377: Could cite some previous studies that use atmospheric $CO_2$ to study NEE over Russia in 2010:

Guerlet, S., Basu, S., Butz, A., Krol, M., Hahne, P., Houweling, S., Hasekamp, O. P., & Aben, I. (2013). Reduced carbon uptake during the 2010 Northern Hemisphere summer from GOSAT. Geophysical Research Letters, 40, 2378–2383. https://doi.org/10.1002/grl.50402

Ishizawa, M., Mabuchi, K., Shirai, T., Inoue, M., Morino, I., Uchino, O., Yoshida, Y., Belikov, D., & Maksyutov, S. (2016). Inter-annual variability of summertime $CO_2$ exchange in Northern Eurasia inferred from GOSAT $XCO_2$. Environmental Research Letters, 11(10), 105001.

Response: Thank you! We have added these two citations in the revised manuscript, see page 15, line 435; page 23, lines 686-689; page 24, lines 713-715.

L434: "on extreme climates" should be "to climate extremes"

Response: We have changed "on extreme climates" to "to climate extremes" in the revised manuscript, see page 17, line 508.

Figure 7 caption: Give the latitude range of these regions in the caption. Rename "low latitudes" to "tropical latitudes" to be consistent with the abbreviation "TL"

Response: Thank you! We have given the latitude range of these regions and renamed "low latitudes" to "tropical latitudes" in the revised manuscript, see page 36, line 992.

Figure 10: It is hard to see the difference because the trends are so large. Please either add panels showing the data-model differences or add a plot to the supplement showing these differences.

Response: Thank you for this suggestion. We have added a plot to the supplement showing these differences (see Figure S11 in the revised supplement).

Supplementary figure S2: This caption is incomplete. What are the individual dots? Different Years?

Response: Thank you! We have changed the caption as "Comparisons between this study and GCP2020 for the estimates of annual (a) NBE and (b) AGR from 2010 to 2019" in the revised supplement.

Reference:

Liu, J., Baskaran, L., Bowman, K., et al.: Carbon Monitoring System Flux Net Biosphere Exchange 2020 (CMS-Flux NBE 2020), Earth Syst. Sci. Data, 13, 299–330, https://doi.org/10.5194/essd-13-299-2021, 2021.

Friedlingstein, P., O'Sullivan, M., Jones, M. W., et al.: Global Carbon Budget 2020, Earth Syst. Sci. Data, 12, 3269–3340, https://doi.org/10.5194/essd-12-3269-2020, 2020.

---

## Author Comment (AC2)

**Referee #2**

Many thanks for the anonymous referee's comprehensive review and valuable suggestions, which help us to present our results more clearly. In response, we have made changes according to the referee's suggestions and replied to all comments point by point. All the page and line number for corrections are referred to the revised manuscript with tracked changes, while the page and line number from original reviews are kept intact.

The authors provide a detailed description of their new GCAS2021 dataset for monthly global NEE data over 10 years from 2010-2019, which were inferred by using their data assimilate system GCASv2 to assimilate the latest GOSAT XCO2 retrievals of version 9. The manuscript is well written, easy to understand while clearly covering important aspects of the dataset, from their inversion method to the comprehensive evaluation using independent data etc. The resulting dataset is meaningful, and easy to use. Together with other top-down and bottom-up NEE /NBE data, it can be used to improve our understanding of terrestrial biosphere-atmosphere carbon exchange, although (just like other NEE/NBE datasets) there are still some open questions on its reliablity, particularly over regions poorly covered by GOSAT observations. Hence, I recommend it for publication after minor correction.

Major comments:
1. As shown in Figure 6, discrepancies from other top-down inversions (such as CMS or CT2019b) are quite significant over tropical regions, and also over South America temperate and South Africa. As mentioned by the authors, it could be caused by different observation coverages. To help the reader understand the impacts of poor (GOSAT) observation coverage over regions (like the Tropical South America etc) on the top-down flux inversion, I suggest the authors include a simple comparison with their own and other groups' inversions based on the denser OCO-2 $XCO_2$ data for 2015-2019. Such comparisons may also help answer the question whether the high net flux in 2019 (Table 1) is realistic.

Response: Many thanks for this suggestion. We have conducted an additional comparison between this study and CMS-Flux for the periods of 2010 to 2014 and 2015 to 2018, respectively, since in the first stage, the $XCO_2$ used in these two studies are almost the same (both GOSAT), while in the second stage, they are different. We found that except for southern Africa, the differences between the two are significantly smaller in 2010-2014 than in 2015-2018 in the above-mentioned regions (Figure R1), confirming that the significant differences are mainly from the different $XCO_2$ products used in these two studies. For southern Africa, we further examine the prior and posterior NBE over southern Africa in these two studies, and find that the prior NBE used in these two systems are quite different (a strong sink in CMS-Flux, and a source in this study). During 2010-2014, the changes relative to the prior NBE constrained by GOSAT are rather small in both studies (Figure R2), resulting in the large difference in

the posterior NBE between these two studies, while in the second stage, because of the better spatial coverage of OCO-2 $XCO_2$, the changes in CMS-Flux increase significantly, resulting in a shift of NBE from a priori strong sink to a posterior medium source, thus reducing the difference in the posterior NBE between these two studies. We have added following sentences in the revised manuscript (see page 11, lines 310-327), and added Figure R1 and R2 in the revised supporting information, which are named as Figure S6 and S7.

"The differences between this study and CMS-Flux NBE 2020 may be related to the different $XCO_2$ products used. As mentioned before, the NBE of CMS-Flux from 2010-2014 and 2015-2018 were inferred from GOSAT and OCO-2 products, respectively. In general, OCO-2 $XCO_2$ has much better spatial coverage than GOSAT $XCO_2$. Wang et al. (2019) pointed out that data amount is one of the most important factors affecting the inversion results, generally, in one region with more $XCO_2$ data, the carbon flux relative to the prior flux is changed more. Therefore, we conduct an additional comparison for the periods of 2010 to 2014 and 2015 to 2018, respectively, since in the first stage, the $XCO_2$ used in these two studies are almost the same (both GOSAT), while in the second stage, they are different. As shown in Figure S6, except for southern Africa, the difference between the two is significantly smaller in 2010-2014 than in 2015-2018, especially in temperate S. America, northern Africa, and Australia, confirming that the significant differences are mainly from the different $XCO_2$ products used in these two studies. In addition to $XCO_2$ data, the prior carbon flux can also have a significant impact on the inversion results (Philip et al., 2019). We further examine the prior and posterior NBE over southern Africa in these two studies, and find that the prior NBE used in these two systems are quite different (a strong sink in CMS-Flux, and a source in this study). In the first stage, the NBE changes ($\Delta_{NBE}$, a posteriori minus a priori) due to the GOSAT constraints are quite small in both studies (Figure S7), resulting in the large difference in the posterior NBE between these two studies, while in the second stage, because of the better spatial coverage of OCO-2 $XCO_2$, the $\Delta_{NBE}$ in CMS-Flux increase significantly, resulting in a shift of NBE from a priori strong sink to a posteriori medium source, thus reducing the difference of the posterior NBE in these two studies. We also find that there is also an increase in the $\Delta_{NBE}$ in this study, which may be related to the increase of GOSAT $XCO_2$ data from 2010 to 2019 (Taylor et al., 2022)."

[Figure]

Figure R1. Comparison of NBE between this study and CMS-Flux NBE 2010 for the periods of 2010-2014 and 2015-2018

[Figure]

Figure R2. Changes in posterior NBE relative to prior fluxes in southern Africa (positive means source increase)

2. I'd like to see more details on the assumption of the a priori error covariance, and like to know how the authors aggregated posterior error across different assimilation windows to calculate the uncertainty for annual flux. Table 1 shows that assimilation of GOSAT XCO2 has reduced the uncertainty of the global annual NEE total by about 16% (i.e., from 0.6 PgC/yr to about 0.5 PgC/yr), which seems lower than other literatures. I am not sure whether the temoral error correlations between assimilation windows have been taken into account in the calculation of those annual uncertainties.

Response: Thank you for this suggestion. We have added more details about the method of calculating regional and global prior and posterior uncertainties in monthly and annual scales in the revised supplement (see Text S1 in the revised supporting information). We have analyzed the uncertainty reduction (UR) in our previous study (Jiang et al., 2021), and compared it with the other studies. The annual mean URs are in the range of 6%–27% across TRANSCOM regions, and the highest monthly

regional UR is about 45 %, which are indeed lower than those given in previous studies. The temporal error correlations between assimilation windows were not considered in the calculation of those annual uncertainties, which may be the main reason for these lower URs. In addition, the shorter DA window (1 week) used in this study may be another reason. In addition to the Text S1 in the revised supplement, the following sentences are also added in the revised manuscript.

Page 6, line 158:

"… uncertainty for NEE and OCN flux about 0.6 and 0.2 PgC yr-1, respectively (for the method, see Text S1)"

Page 8, lines 222-223:

"… We also provide a Fortran program for the calculation of posterior uncertainties. The method for calculating posterior uncertainties is given in the Text S1 in the Supporting Information."

Minor comments:
1. Line 31, Page 1: 'We believe that this dataset will contribute to regional or national-scale carbon cycle and carbon neutrality assessment …'. I think this dataset can be very useful, particularly when combined with other top-down and bottom-up results. But for me, the above statement is a bit too strong, considering the significant discrepancies with other datasets over several regions critical for global carbon cycle. I also don't see direct comparisons/evaluation at national scale. I think further assessment are needed, and at this moment, it is better to say ''this dataset can contribute to …'

Response: Many thanks for this comment and suggestion. We agree with you that this statement is a bit too strong, because the comparisons and evaluations at national scale are not performed in this study. We have modified that sentence to "this dataset can contribute to …" in the revised manuscript, see page 1, line 31.

2. Line 73, Page 3:  'data are now available'. Change to 'data is now available…'  (to be consistent with line 74 ', which spans…' )

Response: Thank you! We have changed 'data are now available' to 'data is now available' in the revised manuscript, see page 3, line 73.

3. Line 135, Page 5: '… the product of CT2017…'. 'CT2017' has not been mentioned before. Should it be 'CT2019B' instead ?

Response: The initial field was indeed obtained from CT2017, not CT2019B. We have changed 'the product of CT2017' to 'the product of CarbonTracker, version 2017 (CT2017)' in the revised manuscript, see page 5, line 141.

4. Line 150, Page 5: Equation 1. Please define $i$ and $N$. It is a bit confusing as the gridded product was at a horizontal resolution of 1x1, but the transport model was run at 1.9x2.5.

Response: Thank you! $i$ is the identifier of the perturbed samples, $N$ is the ensemble size. For the horizontal resolution, indeed, the spatial resolution of the optimized flux cannot be higher than that of the atmospheric transport model. In our EnSRF assimilation algorithm, the spatial resolution of the perturbation factor we adopted is 3°×3°, and the resolution of the prior fluxes is 1°×1°, that is, the prior fluxes within each 3° grid have the same perturbation factor. We have added these descriptions in the revised manuscript.

Pages 5-6, lines 152-153:

"$i$ is the identifier of the perturbed samples, N is the ensemble size (here 50)."

Page 6, lines 155-156:

"… the prior fluxes of NEE, FIRE, FFC and OCN, respectively. The spatial resolution of the perturbation factor ($\delta_i×\lambda$) we adopted is 3°×3°, and the resolution of the prior fluxes is 1°×1°, that is, the prior fluxes within each 3° grid have the same perturbation factor. In each 3° grid …"

5. Line 239, Page 9: '…which also shown'. Should be '…which also showed'

Response: Thank you! We have changed '…which also shown' to '…which also showed' in the revised manuscript, see page 10, line 270.

6. Line 251, Page 9: '…shown that'. Should be '...showed that '

Response: Thank you! We have changed '… shown than' to '… showed that' in the revised manuscript, see page 10, line 282.

7. Line 262, Page 9 & Figure 6 Caption. Please use 'TRANSCOM' or 'TRANSCOM 3' consistently

Response: We have changed the captions of Figure 3 and Figure 6 to use 'TRANSCOM' consistently in the revised manuscript, see page 34, line 979, and page 36, line 988.

8. Line 315, Page 8 & Table 1. It is better to include prior estimates for comparison. Also, why were the CMS and CT2019B results not included in this table?

Response: Many thanks for this suggestion. In this manuscript, we have placed all tables and figures on posterior estimates in the main text, and graphs comparing prior and posterior estimates in the supporting information. Since we have presented a comparison plot of the prior and posterior global NEE in Fig. S6l (Figure S9l in the revised supplement), we do not include the prior estimates in Table 1. However, we have added a description about the comparison between the prior and posterior NEE in that paragraph, and in addition, following this suggestion, we have also included the estimates of CMS-Flux NBE 2020 and CT2019B in Table 1, and added corresponding descriptions in that paragraph in the revised manuscript (see pages 8-9, lines 232-242; page 41, lines 1042-1045). The added descriptions are as follows:

"Compared with the prior NEE (Figure S9l), the posterior NEEs increase significantly from 2010 to 2012, and decrease to varying degrees (in range of 0.15 to 1.15 PgC yr$^{-1}$) from 2015 to 2019. Table 1 also lists the estimates from the CMS-Flux (CMS-Flux NBE 2020, Liu et al., 2021) and CarbonTracker (CT2019B, Jacobson et al., 2020) systems. CMS-Flux NBE 2020 is a product for the period of 2010-2018, in which the results of 2010-2014 was inverted from the GOSAT XCO$_2$ v7.3, and the rests were inverted from the OCO-2 XCO$_2$ v9 retrievals. Both GOSAT and OCO-2 retrievals were from the ACOS team, created using the same retrieval algorithm and validated using the same strategy (Liu et al., 2021). CT2019B is a product inverted from global surface, tower and aircraft CO$_2$ measurements. CMS-Flux NBE 2020 only presented the NBE results, and the FIRE emission used in this study and CT2019B are also different. Therefore, this comparison focuses on NBE. In 2010 and 2014, our estimates are close to CT2019B and significantly lower than the estimates of CMS-Flux NBE 2020; in contrast, in 2011, 2012, 2013, 2016 and 2017, they are comparable to CMS-Flux NBE 2020 and higher than those of CT2019B. In 2015, it is higher than both."

9. Table 1: Table 1 shows a high net flux of 6.08 PgC/yr during 2019, which is higher than 2015 (5.95 PgC/yr). It seems inconsistent with NOAA atmospheric CO2 growth rates derived from the in-situ network (i.e., 2.57 ppm/yr (2019) vs 2.96 ppm/yr (2015)). Also, to my knowledge, some inversions based on OCO-2 XCO2 data or based on the surface insitu network showed significantly lower net global fluxes (up to 1 PgC/yr). Some discussions may be needed.

Response: Thank you for this suggestion. In our inversion, the inverted net flux in 2019 is indeed higher than that in 2015, and higher than the AGR in 2019 observed by NOAA, which is mainly due to the abnormally low carbon sink in the tropical latitudes (TL, 30° S ~ 30° N) in 2019 (Figure 7). The reason may be related to the XCO₂ retrievals of GOSAT. After detrending, the GOSAT XCO₂ in 2019 is higher than that in 2015, while OCO-2 is the opposite (Figure R3). The following sentences have been added in the revised manuscript (see page 9, lines 254-259), and the Figure R3 has been added in the revised supplement, and named as Figure S3.

"It also should be noted that in this study, the AGR in 2019 is higher than that in 2015, and significantly higher than the observed value, which is mainly due to the abnormally low carbon sink in the tropical latitudes (TL, 30° S ~ 30° N) in 2019 (Figure 7). The reason may be related to the biases in the GOSAT XCO₂ retrievals in TL. We analyze the monthly changes of GOSAT XCO₂ in 2015 and 2019, and compare them with the OCO-2 XCO₂ retrievals (OCO-2 v10). We find that after detrending, in TL, the GOSAT XCO₂ in 2019 is higher than that in 2015, while OCO-2 is the opposite (Figure S3)."

[Figure]

Figure R3. Monthly variations of (a) XCO₂ and (b) NBE in tropical latitudes (TL, 30° S ~ 30° N) in 2015 and 2019 (because GOSAT lacks data in January 2015, in each year, XCO₂ for each month is its change relative to February. It could be found that the carbon sinks in January-August and September-December 2019 were significantly smaller and stronger than those in the same period in 2015, respectively. Correspondingly, compared with 2015, GOSAT has higher XCO2 in March - August, and lower ones in September-December in 2019. Although OCO-2 has a similar pattern, compared with 2015, the XCO₂ increase in March-August is significantly smaller than that of GOSAT, while the decrease in September-December is significantly higher than that of GOSAT. On average, the GOSAT XCO2 in 2019 is higher than that in 2015, while OCO-2 is the opposite)

10. Line 425, Page 15: ' In the Amazon basin, the simulated, $CO_2$ profiles also agree well with the observations...'. I think the results (Figure 12) actually suggest there could be systematic bias in the posterior fluxes over Amazon basin.

Response: Thank you for this suggestion. Indeed, that conclusion is too arbitrary. We re-analyzed the comparison results of simulation and observation in the Amazon region, and found that there is negative BIAS (~ - 1.0 ppm) below 1 km, small BIAS (~ 0.2 ppm) at 1 ~ 1.5 km and positive BIAS (~ 0.9 ppm) above 1.5km. This indicates that there is indeed a systematic bias in the posterior fluxes over Amazon basin, which may be caused by the model vertical transport error. We have reorganized those sentences in the revised manuscript (see page 17, lines 494-501), which are shown as follows:

"In the Amazon basin, the MAE and RMAE of all 4 sites decrease with height, with MAE and RMSE decreasing from about 2 ppm near 1000 m height to about 1.5 ppm near 4000 m. For BIAS, below 2000 m, they increase significantly with height. There are negative (~ -1.0 ppm, data not shown), small (~ 0.2 ppm), and significant positive BIAS (~ 0.9 ppm) below 1000 km, at 1000 ~ 1500 m, and 1500 ~ 2000 m heights, respectively, indicating that there are considerable vertical transport errors, and the carbon sinks over tropical S. America may have systematic biases."

Reference:

Jiang, F., Wang, H., Chen, J. M., et al.: Regional $CO_2$ fluxes from 2010 to 2015 inferred from GOSAT $XCO_2$ retrievals using a new version of the Global Carbon Assimilation System, Atmos. Chem. Phys., 21, 1963–1985, https://doi.org/10.5194/acp-21-1963-2021, 2021.

---

## Author Comment (AC3)

**Referee #3**

Many thanks for your comprehensive review and valuable suggestions, which help us to present our results more clearly. In response, we have made changes according to the referee's suggestions and replied to all comments point by point. All the page and line number for corrections are referred to the revised manuscript with tracked changes, while the page and line number from original reviews are kept intact.

**Comments for essd-2022-15**

This manuscript introduces a new global top-down NEE data from 2010 and 2019 produced by GCASv2 assimilating with a new ACOS GOSAT XCO2 L2 data. The data has been well validated by ground based and aircraft in-situ measurement that provide from OBSPACKv6. Further studies on comparison with GCP data descripts clear in this manuscript. The detail of dataset, e.g. interannual variations, trend, and profile performance is introduced. This manuscript, overall, is well written and organized, and detail is enough for potential readers and data users. I recommend it for publication, but a minor revision is required.

Line 100, the griding method is not clear, would suggest adding some equation description.

Response: Thank you for this suggestion. We have given more details about the griding method in the revised manuscript (see page 4, lines 100-107), which are shown as follows.

"... we re-grid the XCO2 data into  $1^{\circ} \times 1^{\circ}$  grid cells. The pixel level XCO2 data are filtered with xco2\_quality\_flag, which is a simple quality flag denoting science quality data (0=Good, 1=Bad), and provided along with the XCO2 product. In each  $1^{\circ} \times 1^{\circ}$  grid and each day, only the XCO2 with xco2\_quality\_flag equals 0 are selected and averaged according to Equation (1).

$$C_{G,T} = \frac{1}{W} \sum_{l=1}^{W} C_{l,t}, \ T = \frac{1}{W} \sum_{l=1}^{W} t$$
(1)

where  $C_{l,t}$  denotes the selected pixel level XCO2 located in grid *G* of one day, *l* is the identifier of the record, *t* is the observation time, and *W* denotes the number of  $C_{l,t}$ . *T* is the averaged observation time, and  $C_{G,T}$  is the re-grided XCO2 concentrations. The other variables in the XCO2 product like column-averaging kernel ..."

Line 155, please state the 'a global ocean circulation and biogeochemistry model'

Response: Thank you! We have added the following sentences in the revised manuscript (see page 4, lines 119-122).

"... Following Jiang et al. (2021), the fluxes in 2009 modeled using a combined global ocean circulation (OPA) and biogeochemistry model (PISCES-T) (Buitenhuis et al.,

2006) is used to fill the no data areas. The sea-air CO2 fluxes simulated using the PISCES-T model have been used in many studies of ocean carbon cycle dynamics (e.g., McKinley et al., 2006; Valsala et al., 2012; Le Quéré et al., 2007), and also used as a priori ocean fluxes in previous inversion studies (e.g., Jiang et al., 2014; Deng et al., 2011; Chen et al., 2017)."

Line 225 and Global carbon budgets, the different between top-down estimation NBE and AGR of GCASv2 and GCP2020 comes from LULUC. I would like to indicate the average significant on this different, otherwise the improvement compared a prior and posterior is not clear.

Response: Thank you for this comment. Since the LULUC carbon emissions have been included in the NBE, the difference in NBE between GCASv2 and GCP2020 mainly comes from the imbalance item of GCP2020. After including the imbalance item in the NBE like Liu et al. (2021), the difference of NBE between this study and GCP2020 could be significantly reduced, especially for the year of 2016. For the difference in AGR, it may be mainly from the biases in the GOSAT XCO2 retrievals. We find that the inverted AGR in 2019 in this study is significantly higher than that of GCP2020, and it is also higher than that in 2015, which is a year with extreme El Niño event. The higher AGR in 2019 is mainly due to the abnormally low carbon sink in the tropical latitudes (TL,  $30^{\circ}$  S ~  $30^{\circ}$  N) in 2019. We find that after detrending, in TL, the GOSAT XCO2 in 2019 is higher than that in 2015, which is unreasonable.

To make it clear, we have added the following sentences in the revised manuscript (see page 9, lines 253-259).

"... The difference in NBE between this study and GCP2020 is partly due to the imbalance item in GCP2020, especially in 2016. It also should be noted that in this study, the AGR in 2019 is higher than that in 2015, and significantly higher than the observed value, which is mainly due to the abnormally low carbon sink in the tropical latitudes (TL,  $30^{\circ}$  S ~  $30^{\circ}$  N) in this year (Figure 7). The reason may be related to the biases in the GOSAT XCO2 retrievals in TL. We analyze the monthly changes of GOSAT XCO2 in 2015 and 2019, and compare them with the OCO-2 XCO2 retrievals (OCO-2 v10). We find that after detrending, in TL, the GOSAT XCO2 in 2019 is higher than that in 2015, while OCO-2 is the opposite (Figure S3)."

Line 237, 'In N. America, the distribution of NEE constraint with GOSAT XCO2 agrees well with a recent regional inversion using surface CO2 and 14CO2 measurements, which also shown significant sources over western US and sinks over central and eastern US (Basu et al., 2020).' Please revise this presentation to avoid over-estimation on the ability of your inversion and GOSAT XCO2 measurement.

Response: Thank you for this suggestion. We have revised this description and changed 'agrees well with' to 'exhibits a similar pattern to that of' in that sentence in the revised manuscript (see page 10, line 269).

Line 265, it seems CMS-Flux using two satellites measurement in their study, incl. GOSAT and OCO-2 comes from different retrieval. The statement of measurement is not clear, e.g. satellite, retrieval algorithm and version, please revise.

Response: Thank you for this suggestion. Both GOSAT and OCO-2 retrievals are from the ACOS team, the versions are v7.3 and v9, respectively. They were created using the same retrieval algorithm and validated using the same strategy (Liu et al., 2021). We have added a sentence to make it clear in the revised manuscript (see page 8, lines 236-237), which is shown as follows.

"...in which the results of 2010-2014 was inverted from the GOSAT XCO2 v7.3, and the rests were inverted from the OCO-2 XCO2 v9 retrievals. Both GOSAT and OCO-2 retrievals were from the ACOS team, created using the same retrieval algorithm and validated using the same strategy (Liu et al., 2021)."

Line 387, what is that 'absolute errors' mean?

Response: Thank you! We have changed 'absolute errors' to 'absolute biases between the posterior CO2 concentrations and CO2 measurements' in the revised manuscript (see page 15, lines 447-448).

Fig.9, this comparison method is not clear.

Response: Many thank for this comment. Yes, the comparison method is not clear here, we have modified the caption of Figure 9 to "Spatial distributions of the (a) BIAS and (b) MAE of the posterior CO2 concentrations at each site (simulations minus observations, unit: ppm)" in the revised manuscript (see page 38, line 1000).

Reference:

Liu, J., Baskaran, L., Bowman, K., et al.: Carbon Monitoring System Flux Net Biosphere Exchange 2020 (CMS-Flux NBE 2020), Earth Syst. Sci. Data, 13, 299–330, https://doi.org/10.5194/essd-13-299-2021, 2021.

---

## Author Response (AR2)

**Comment**: I thank the reviewers for addressing all of my comments. I'm a little perplexed by the trend in bias against in situ data. The fact that a very similar trend is found when estimating the sink from the fluxes and when comparing the $CO_2$ fields against the in situ $CO_2$ measurements suggests that the trend is a result of either (1) a $CO_2$ growth rate trend in ACOS v9 GOSAT that is inconsistent with the in situ $CO_2$ measurements or (2) a residual trend in the inversions fit to the GOSAT data. I think that it would be worthwhile to determine which of these factors is driving this. I would encourage the authors to add a timeseries plot of the mismatch between the posterior $CO_2$ fields and GOSAT ACOS v9 $XCO_2$ data to see if there is a relative trend similar to the in situ $CO_2$ data used for validation.

**Response:** Many thanks for this suggestion, which is really helpful for the analysis of the inversion results in this study and the improvement of our system in the future. We analyzed the timeseries of the global averaged monthly mean posterior $XCO_2$ and GOSAT $XCO_2$ concentrations. As shown in Figure 1, the mismatches between the posterior $XCO_2$ fields and GOSAT ACOS v9 $XCO_2$ data also have an increasing trend from 2010 to 2019, with an annual mean increment about 0.09 ppm yr$^{-1}$, indicating that the trend in bias against in situ data is a result of a residual trend in the inversions fit to the GOSAT data.

We have added the following sentences in the revised manuscript (see lines 472-475, page 16) and added Figure 1 in the revised Supporting Information as Figure S13.

"… On global average (74 sites), the annual mean biases increase from -0.36 ppm in 2010 to 0.75 ppm in 2019, with uptrend slope of 0.115 ppm yr$^{-1}$ (Figure S12). By multiplying by a factor of 2.124 PgC ppm$^{-1}$ (Ballantyne et al., 2012), this bias accumulation rate is equal to 0.244 PgC yr$^{-1}$, which is very consistent with the 10-year averaged bias in the inverted global AGR given in Section 5.1 (0.25 PgC yr$^{-1}$). This uptrend is a result of a residual trend in the inversions fit to the GOSAT data. We analyzed the timeseries of the global averaged monthly mean posterior $XCO_2$ and GOSAT $XCO_2$ concentrations, and found that the mismatches between the posterior $XCO_2$ fields and GOSAT data also have an upward trend from 2010 to 2019, with an annual mean increment about 0.09 ppm yr$^{-1}$ (Figure S13)."

[Figure]

**Figure 1** Global mean monthly $XCO_2$ from 2010 to 2019 (the small figure shows the annual mean biases and bias increment in each year)